# Proximity superconductivity in atom-by-atom crafted quantum dots

Lucas Schneider[1✉], Khai That Ton[1], Ioannis Ioannidis[2,3], Jannis Neuhaus-Steinmetz[1], Thore Posske[2,3], Roland Wiesendanger[1] & Jens Wiebe[1]

Gapless materials in electronic contact with superconductors acquire proximity-induced superconductivity in a region near the interface[1,2]. Numerous proposals build on this addition of electron pairing to originally non-superconducting systems and predict intriguing phases of matter, including topological[3–7], odd-frequency[8], nodal-point[9] or Fulde–Ferrell–Larkin–Ovchinnikov[10] superconductivity. Here we investigate the most miniature example of the proximity effect on only a single spin-degenerate quantum level of a surface state confined in a quantum corral[11] on a superconducting substrate, built atom by atom by a scanning tunnelling microscope. Whenever an eigenmode of the corral is pitched close to the Fermi energy by adjusting the size of the corral, a pair of particle–hole symmetric states enters the gap of the superconductor. We identify these as spin-degenerate Andreev bound states theoretically predicted 50 years ago by Machida and Shibata[12], which had—so far—eluded detection by tunnel spectroscopy but were recently shown to be relevant for transmon qubit devices[13,14]. We further find that the observed anticrossings of the in-gap states are a measure of proximity-induced pairing in the eigenmodes of the quantum corral. Our results have direct consequences on the interpretation of impurity-induced in-gap states in superconductors, corroborate concepts to induce superconductivity into surface states and further pave the way towards superconducting artificial lattices.

Particularly interesting states of matter are formed when superconductivity is induced into intrinsically non-superconducting materials by the proximity effect[1,2] based on Andreev reflection processes at the interface. If the transparency of the interface between a normal metal in the clean limit and the superconductor is high, superconductivity is induced over a length scale that can exceed dozens of nanometres[15]. However, for many heterostructures, superconductivity has to be induced through interface states or into surface states[6,16,17]. These are typically well decoupled from the bulk bands and, thus, it is unclear a priori whether they acquire sufficient pairing if their distance to the superconductor is larger than a few nanometres[15–17]. To study this effect in detail, we downscale the problem as much as possible by investigating only a single resonance mode of a surface state. This is achieved by laterally confining the surface state in a quantum corral, forming a particular quantum dot (QD). These can naturally occur in nanoscopic islands[18,19] or, in a more tunable platform, in artificially designed adsorbate arrays[11,20], in which the QD walls are built atom by atom using the tip of a scanning tunnelling microscope as a tool. Although the surface states are typically well decoupled from metallic bulk states in the direction perpendicular to the surface plane, scattering at step edges or the adsorbates is known to introduce a measurable coupling to the bulk electronic states, leading to a lifetime broadening of the QD's eigenmodes $\Gamma$ on the order of several meV (refs. 21,22). Notably, in contrast to the usual cases of the more widely studied semiconductor or molecular

QDs[23], the electron density screening the metallic QDs investigated here is by orders of magnitude larger, which leads to largely suppressed electron–electron interactions, that is, the QD charging energy $U$ is negligible and, thereby, the QD can be described by spin-degenerate single-particle eigenmodes. Coupled arrays of such QDs with tunable interactions between adjacent sites have evolved as an exciting platform for the simulation of quantum materials[24,25]. However, although there has been progress in choosing different material templates for incorporating more complex phenomena such as, for example, Rashba spin–orbit coupling into these QDs[26], pathways for inducing superconductivity into their individual eigenmodes have not been studied so far.

Here we investigate artificial QDs defined by a cage of Ag atoms on thin Ag(111) islands (see Fig. 1a,b and Methods) grown on superconducting Nb(110) using scanning tunnelling microscopy (STM) and scanning tunnelling spectroscopy. We use superconducting Nb tips, leading to enhanced energy resolution and a shift of spectral features to higher energies by the value of the tip's superconducting gap $\Delta_\mathrm{t}$, that is, states at the sample's Fermi energy $E_\mathrm{F}$ are found at bias voltages of $eV = \pm\Delta_\mathrm{t}$ (Supplementary Note 1). The proximity to Nb(110) opens a superconducting gap of $2\Delta_\mathrm{s} = 2.70$ meV in the bulk states of Ag(111) for island thicknesses well below $d_\mathrm{Ag} = 100$ nm (refs. 15,27) (see Methods). The outline of the experiment is shown in Fig. 1b: the scattered Ag(111) surface-state electrons visible as wavy patterns at the surface of Ag islands (Fig. 1a) are confined within a couple of lattice constants in the

[1]Department of Physics, Universität Hamburg, Hamburg, Germany. [2]I. Institute for Theoretical Physics, Universität Hamburg, Hamburg, Germany. [3]The Hamburg Centre for Ultrafast Imaging, Hamburg, Germany. ✉e-mail: lucas.schneider@physnet.uni-hamburg.de

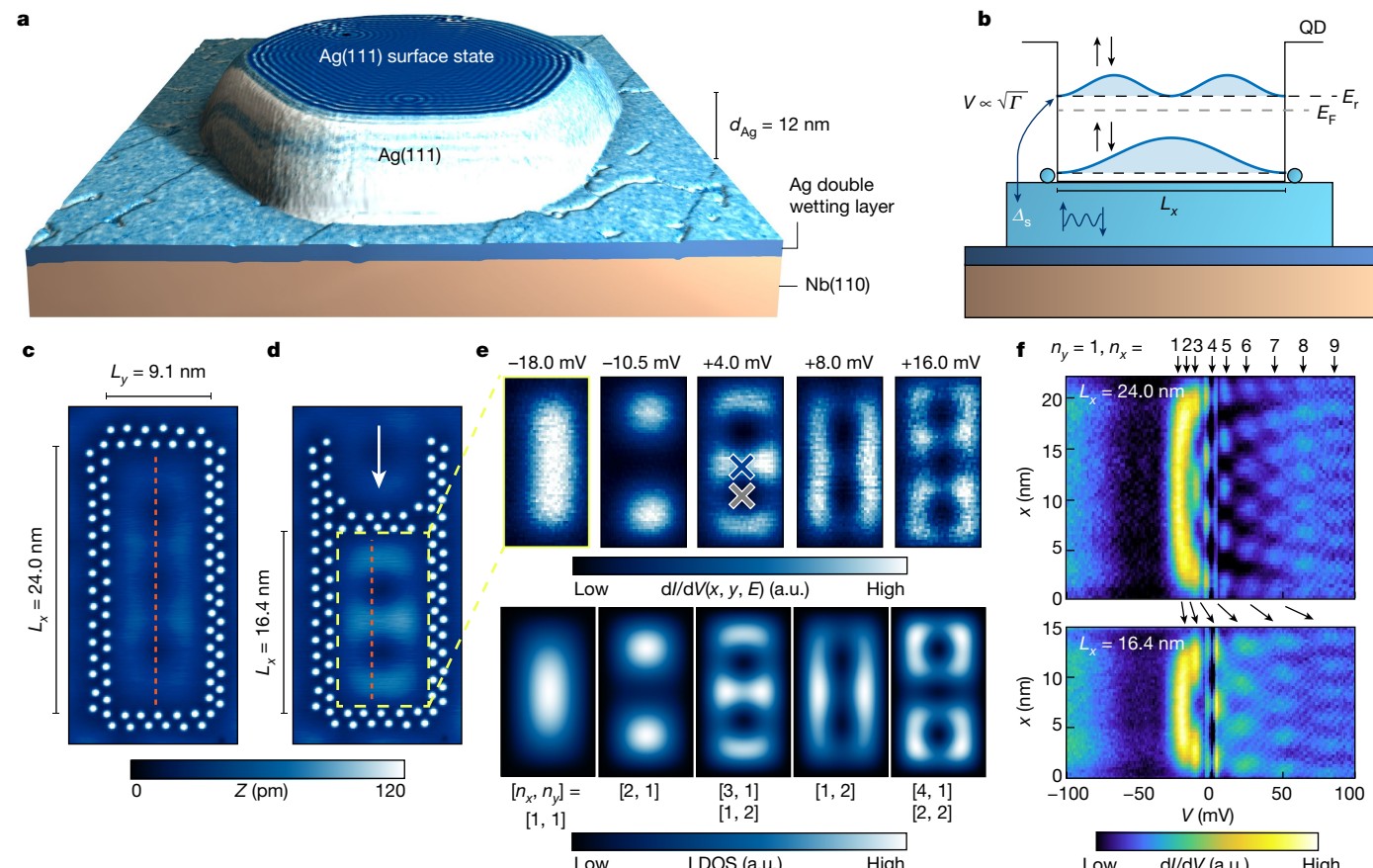

**Fig. 1 | Atom-by-atom built QDs coupled to a superconducting substrate.**
**a**, Three-dimensional rendering of the constant-current STM topography of a Ag island with a thickness of 12 nm. The simultaneously measured d$I$/d$V$ signal is used as the texture of the model. The island grows on top of a pseudomorphic Ag double layer on Nb(110) (sketched profile; see Methods). **b**, Sketch of the experimental setup with the QD walls laterally confining the surface-state electrons into spin-degenerate QD eigenmodes of energies $E_r$. The eigenmodes couple to the superconducting substrate ($\Delta_s$) with a strength $V \propto \sqrt{\Gamma}$. $E_r$ can be pitched by adjusting the width $L_x$ of the QD. **c**, Constant-current STM image of a rectangular QD with side lengths $L_x$ and $L_y$ consisting of 44 Ag atoms. $L_x$ and $L_y$ are defined as the distance between the Ag atoms in the inner ring. $Z$, apparent height. **d**, Constant-current STM image of the same structure with one of the QD walls moved as indicated by the arrow. **e**, Upper panels, constant-height d$I$/d$V$ maps at bias voltages indicated in the respective panels measured in the interior of the QD in panel **d** (area marked by the dashed yellow lines). All panels are 15 × 7.5 nm² in size. Lower panels, simulation of a hard-wall rectangular box with dimensions $L_x$ = 16.4 nm, $L_y$ = 9.1 nm assuming a parabolic dispersion of the quasiparticles with $m_{eff}$ = 0.58$m_e$ and $E_0$ = −26.4 meV (see Methods). The quantum numbers [$n_x$, $n_y$] of the dominant eigenmodes at the energies of the experimental maps (corrected by an offset of $\Delta_{tip}$) are given below each map. **f**, d$I$/d$V$ line profiles along the dashed orange vertical lines marked in panels **c** and **d**. QD eigenmodes with $n_y$ = 1 and $n_x$ as indicated by the arrows at the top are observed. Their respective energy is shifted when the length $L_x$ is altered as illustrated by the black arrows. a.u., arbitrary units.

direction perpendicular to the surface[28] but still have a finite coupling $V \propto \sqrt{\Gamma}$ to the superconducting Ag bulk electrons[22,29]. We further confine these electrons laterally within QDs built of walls of Ag atoms resulting in spin-degenerate eigenmodes of energies $E_r$, which can be pitched to $E_F$ by adjusting the width $L_x$ of the QD. We then investigate the proximity effect of the bulk electrons onto these QD eigenmodes. Note that, owing to the negligible electron–electron interaction energy $U$, we are operating in the regime $U \ll \Delta_s \approx \Gamma \ll \delta E_r$ ($\delta E_r$ is the energetical separation of the QD eigenmodes), which—for semiconductor systems—has only recently been realized in transmon qubits based on superconductor–semiconductor QD–superconductor Josephson junctions[13,14].

Individual Ag atoms (see Methods for details) can be arranged to form rectangular artificial QDs of tunable sizes (Fig. 1c,d) using lateral atom-manipulation techniques (see Methods). The spatial structure of the QD's eigenmodes can be mapped by measuring the differential conductance d$I$/d$V$($x$, $y$, $E$) at a particular bias voltage $eV = E$. The resulting patterns (Fig. 1e, upper panels) closely resemble the eigenmodes of a two-dimensional rectangular box potential with infinite walls having a well-defined number of antinodes in the $x$ and $y$ directions [$n_x$, $n_y$]

(Fig. 1e, lower panels; see Methods for details). In the following, the width $L_y$ of the QD is kept fixed, whereas the length $L_x$ is tuned by moving the upper Ag wall laterally (see Fig. 1d). This leads to a change in the confinement conditions such that the eigenenergies of the QD states are shifted. Experimentally, this can be verified by measuring d$I$/d$V$ line profiles along lines close to the central axis of a given QD (Fig. 1f, upper panel): the eigenmodes with $n_y$ = 1 and $n_x$ = 1, 2, 3… can be identified and are marked by black arrows. When the QD length $L_x$ is changed from 24.0 nm to 16.4 nm (lower panel; see Extended Data Fig. 1 for a complete set of line profiles measured on QDs with $L_x$ = 3.0 nm to 24 nm), a shift of the individual states to higher energies can be observed (black arrows)[20]. Note that it can already be seen by comparison of the top and bottom panels of Fig. 1f that, by decreasing the length $L_x$ of the QD, the linewidth $\Gamma$ of the eigenmodes and thereby their coupling $V \propto \sqrt{\Gamma}$ to the bulk superconducting electrons increases, which is a well-known effect owing to increased surface–bulk scattering[22,29]. These effects are used in the following to continuously pitch QD eigenmodes with different couplings $V$ through $E_F$ by accordingly tuning $L_x$.

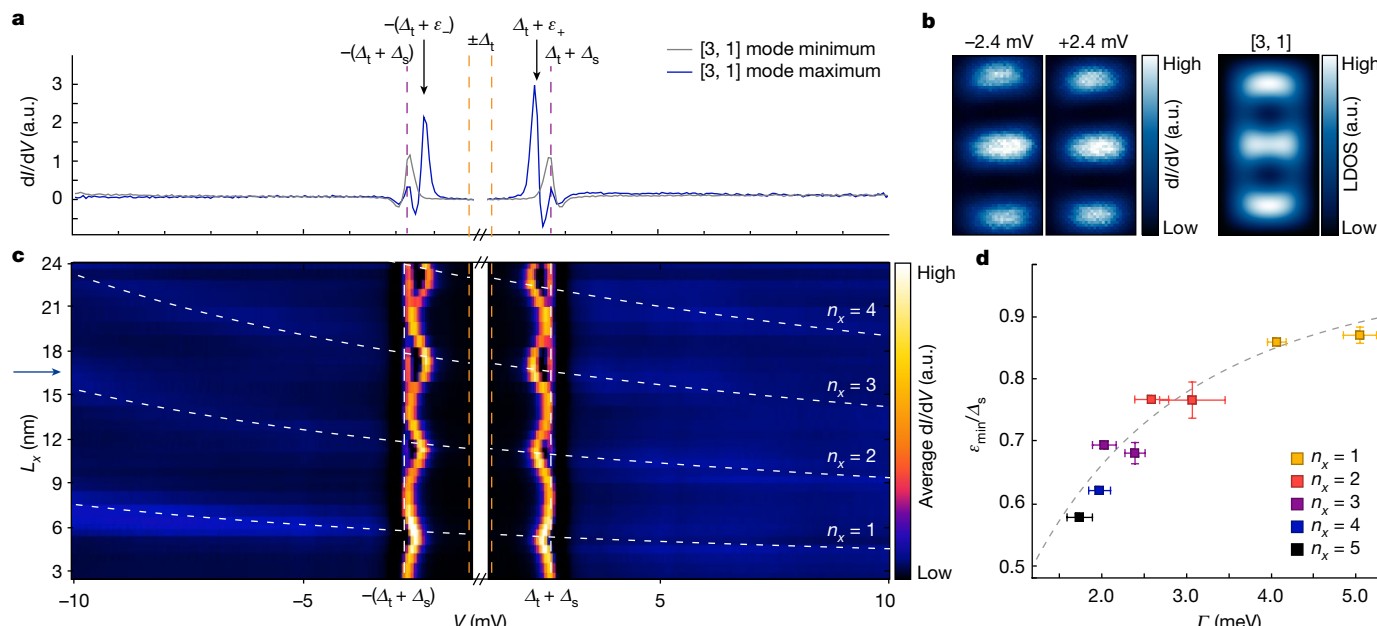

**Fig. 2 | In-gap states of near-zero-energy pitched QD eigenmodes. a**, d$I$/d$V$ spectra measured at two different positions (grey and blue crosses in Fig. 1e) in the QD shown in Fig. 1d–f. The values of the tip's superconducting gap $eV = \pm\Delta_t$ and the sum $eV = \pm(\Delta_t + \Delta_s)$ with the proximity-induced Ag bulk gap $\Delta_s$ are marked by dashed orange and purple lines, respectively. In-gap states appear at energies $\pm(\Delta_t + \varepsilon_\pm)$, marked by black arrows. **b**, Left, constant-height d$I$/d$V$ maps measured at the energies of the in-gap state peaks in the same area as in Fig. 1e. Right, particle-in-a-box simulation evaluated at zero energy with dominant contribution of the eigenmode with $[n_x, n_y] = [3, 1]$. **c**, Evolution of averaged d$I$/d$V$ spectra from d$I$/d$V$ line profiles measured along the central vertical axis of different QDs (see dashed orange lines in Fig. 1c,d) as a function of $L_x$. The dashed white lines mark the evolution of the eigenmodes with $n_y = 1$ and

$n_x = \{1, 2, 3, 4\}$ obtained from fitting the d$I$/d$V$ spectra at energies outside the gap (see Supplementary Note 2). The length of the QD presented in panels **a** and **b** is marked by the blue arrow on the left side. **d**, Linewidths $\Gamma$ of different QD eigenmodes extracted from fitting data from different QDs to Lorentzian peaks at energies outside the gap (see Supplementary Note 2). These are compared with the minimal energies of the in-gap states found when $E_r \approx 0$ (error bars are standard deviations extracted from fitting the data; see Supplementary Note 2 for details). The dashed grey line is the expected theoretical relation for a spin-degenerate level coupled to a superconducting bath[12] (based on equation (13) in Methods). Data on further QDs constructed and analysed as described in Supplementary Note 3 are included in panel **d**. a.u., arbitrary units.

At low energies, d$I$/d$V$ spectroscopy of the QD presented in Fig. 1d shows clean superconductor–insulator–superconductor (SIS) tunnelling without any in-gap states at spatial locations at which no QD eigenmodes are present (grey curve in Fig. 2a and grey cross in Fig. 1e): sharp and prominent peaks appear at bias voltages corresponding to $eV = \pm(\Delta_t + \Delta_s)$, indicating tunnelling between the coherence peaks of tip and sample (the bias range $|eV| < \Delta_t$ is left out in Fig. 2a,c; see Methods and Supplementary Note 1 for more details). The absence of conductance at lower energies confirms that the bulk gap of Ag(111) is fully developed. By contrast, when measuring on a maximum of the QD eigenmode closest to $E_F$, we find a pair of sharp electronic states at particle–hole symmetric energies $\pm(\Delta_t + \varepsilon_\pm)$ within the gap (blue curve in Fig. 2a and blue cross in Fig. 1e). When mapping the spatial distribution of these states (Fig. 2b), we find that they closely resemble the shape of the expected QD eigenmode at $E \approx E_F$ as obtained from particle-in-a-box simulations (rightmost panel). To gain more insight into the nature of these in-gap states, we tune the length $L_x$ of the QD and study the evolution of both the eigenmodes outside and inside the gap (Fig. 2c; see Extended Data Fig. 1 for the full datasets and Supplementary Note 3 for another QD example). As expected, the eigenmodes with quantum numbers $[n_x, 1]$ outside the gap move in energy following the well-known $L_x^{-2}$ behaviour (dashed white lines; see also Supplementary Note 2). Moreover, it can be seen that the peaks at $\pm(\Delta_t + \Delta_s)$ (dashed white vertical lines) remain at the same energy for all QD sizes, indicating that they stem from the proximitized Ag bulk states. Most notably, it can be observed that the in-gap states at varying energies $\pm(\Delta_t + \varepsilon_\pm)$ appear whenever a QD eigenmode energy $E_r$ approaches $E_F$. The absolute value for $\varepsilon_\pm$ is lowest when the length $L_x$ of the QD is such that the $E_r$ would cross $E_F$ if extrapolated from outside the superconducting gap

to the energetical region inside the gap (see dashed lines in Fig. 2c). We evaluate this minimum value $\varepsilon_{min}$ for different eigenmodes of the QD and compare the results with their estimated energetic broadening $\Gamma$ at energies outside the gap (see Supplementary Note 2 for details on the analysis). The energetic broadening is known to be predominantly related to the inverse lifetime of quasiparticles in the respective QD eigenmode for energies close to $E_F$. Furthermore, as noted above, $\Gamma$ of the eigenmodes close to $E_F$ decreases with increased QD size[22,29]. Indeed, this trend can be seen in Fig. 2d for the eigenmodes with increasing $n_x$, that is, for wider QDs. As a main result of this work, there is a clear correlation between $\varepsilon_{min}$ and $\Gamma \propto V^2$: for increased couplings $\Gamma$ of a zero-energy QD eigenmode to the substrate superconductor, $\varepsilon_{min}$ is shifted from $E_F$ towards the gap edge $\Delta_s$ of the substrate (see Fig. 2d).

The observation of these in-gap states in an STM experiment is a surprising result, as impurity-induced states at particle–hole-symmetric energies deep inside the gap of an $s$-wave superconductor are commonly believed to only appear for magnetic impurities[30,31]. In-gap states emerging around non-magnetic impurities are mostly considered to be evidence for unconventional superconductivity[32,33]. In our samples, we exclude that magnetism plays a role on the pure and well-characterized noble-metal surface with only non-magnetic adatoms. Furthermore, Nb is a conventional $s$-wave superconductor and the proximity effect induced in a normal metal with negligible spin–orbit coupling is not expected to induce considerable unconventional pairing. However, as shown theoretically by Machida and Shibata[12] in 1972, there is always a subgap solution for the problem of a localized spin-degenerate level, as present in the QDs in our samples, coupled to a superconducting bath owing to resonance scattering[12,30]. We consider the Hamiltonian of ref. 12, that is,

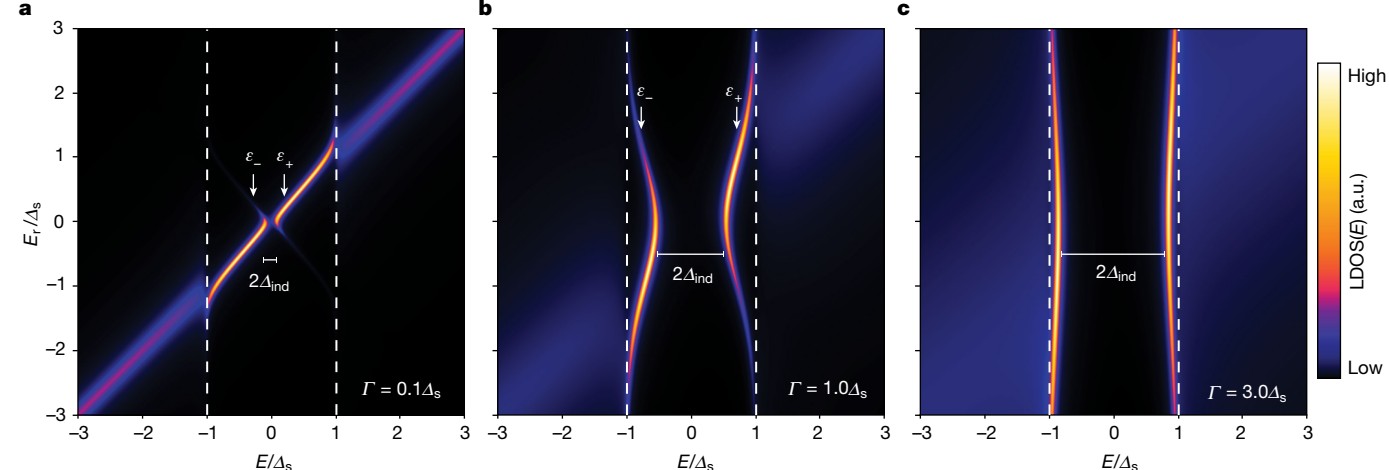

**Fig. 3 | MSSs from resonance scattering at a spin-degenerate localized level.**
**a**, Energy-dependent local electron density of states LDOS($E$) of a single localized level at energy $E_r$ coupled to a superconducting bath with the parameter $\Delta_s$. The coupling strength $\Gamma \propto V^2$ (see Methods) equals $0.1\Delta_s$. The induced gap $\Delta_{ind}$ and the energies of the in-gap states $\varepsilon_\pm$ are marked. **b**, Same as panel **a** but for $\Gamma = 1.0\Delta_s$. **c**, Same as panel **a** but for $\Gamma = 3.0\Delta_s$. An energetic broadening of $\delta E = 0.03\Delta_s$ has been added in all panels (see Methods). a.u., arbitrary units.

$$\mathcal{H} = \sum_{\mathbf{k},\sigma} \epsilon_{\mathbf{k}} c^\dagger_{\mathbf{k},\sigma} c_{\mathbf{k},\sigma} + \sum_{\mathbf{k},\sigma} V(c^\dagger_{\mathbf{k},\sigma} d_\sigma + d^\dagger_\sigma c_{\mathbf{k},\sigma}) + \sum_\sigma E_r d^\dagger_\sigma d_\sigma$$
$$- \Delta_s \sum_{\mathbf{k}} (c^\dagger_{\mathbf{k},\uparrow} c^\dagger_{-\mathbf{k},\downarrow} + c_{-\mathbf{k},\downarrow} c_{\mathbf{k},\uparrow}), \tag{1}$$

in which $c_{\mathbf{k},\sigma}$ ($c^\dagger_{\mathbf{k},\sigma}$) and $d_\sigma$ ($d^\dagger_\sigma$) refer to the annihilation (creation) operators of superconducting bath electrons and the localized level with spin $\sigma \in \{\uparrow, \downarrow\}$, respectively. $\epsilon_{\mathbf{k}}$ denotes the normal electronic dispersion of the superconductor, $V \propto \sqrt{\Gamma}$ is the coupling strength of the localized level at energy $E_r$ to the bath and $\Delta_s$ is the order parameter of $s$-wave superconductivity in the bath. Calculating the local density of states (LDOS) of the level using Green's function equations of motion (see Methods) confirms that there is always a pair of Andreev bound states at in-gap energies[12] for all non-vanishing $V$ (see also Extended Data Fig. 2 and refs. 13,14). In the following, we refer to these states as Machida–Shibata states (MSSs)[12], which are a special kind of Andreev bound states in the limit of negligible electron–electron interactions ($U$). We depict the energy evolution of the MSSs as a function of the localized level's energy $E_r$ in the normal state for different choices of $\Gamma$ in Fig. 3. For $\Gamma/\Delta_s \ll 1$ (Fig. 3a), the localized level couples only weakly to the superconductor and its energy $\varepsilon$ evolves continuously through the gap, whereas its particle–hole-symmetric partner state at $-\varepsilon$ features negligible spectral weight in the LDOS. As $\Gamma/\Delta_s$ is increased (Fig. 3b), the states with $\varepsilon_\pm$ show a pronounced anticrossing behaviour as $E_r$ approaches zero. Moreover, both states at $\varepsilon_\pm$ acquire a finite spectral weight in the LDOS, indicating that the superconductor mixes particle-like and hole-like states. This situation is closely reminiscent of the experimental data in Fig. 2c. For strong coupling $\Gamma/\Delta_s \gg 1$, the in-gap states shift close to $\Delta_s$ irrespective of $E_r$, consistent with the regular proximity effect being induced into the localized resonance level, leading to a full superconducting gap (see Extended Data Figs. 2 and 3 for more detailed simulations of this model and its comparison with experimental data). We observe a similar effect in a tight-binding description of a QD weakly coupled to a superconducting surface layer (Supplementary Note 4), corroborating that the simplified description of the QD's eigenmode as a single localized quantum level $E_r$ shown in Fig. 3 is appropriate. The predicted shift of the minimal energy of the MSS with increasing $\Gamma$ is included as a grey dashed line in Fig. 2d. Its good quantitative agreement with the experimental data without further fitting parameters suggests that the resonances found experimentally are indeed MSSs.

Although these results demonstrate that the lowest-energy quasiparticle excitations of the local level become gradually gapped out with increasing coupling to the superconducting bath, it is not clear a priori whether the local level experiences proximity superconductivity. To this end, we perform a Schrieffer–Wolff transformation of equation (1) to obtain the effective low-energy theory of the level when $E_r$ lies within the gap of the superconductor (see Methods for details). The resulting Hamiltonian reads

$$\mathcal{H}'_D = \sum_\sigma E_r(1 - \Delta_{ind}/\Delta_s) d^\dagger_\sigma d_\sigma - \Delta_{ind}(d^\dagger_\uparrow d^\dagger_\downarrow + d_\downarrow d_\uparrow). \tag{2}$$

Indeed, equation (2) includes a term for the induced pairing energy $\Delta_{ind}$ of the level's quasiparticles, resembling the Bardeen–Cooper–Schrieffer-like mean-field expression for superconductivity. On the basis of equation (2), it can be seen that, for $E_r = 0$, the lowest-energy eigenstates of the system are energetically located at $\varepsilon_\pm = \pm\Delta_{ind}$. Thereby, for negligible electron–electron interactions $U = 0$, the values of $\varepsilon_{min}$ we measured for the different QD eigenmodes (Fig. 2d) can indeed be identified with the proximity-gap magnitudes $\Delta_{ind}$, which approach $\Delta_s$ for strong coupling $\Gamma$.

Notably, the observed in-gap states at $\varepsilon_+$ and $\varepsilon_-$ are not symmetric in intensity. Their peak asymmetry in spectral weight can be analysed in terms of the Bogoliubov mixing angle

$$\theta_B = \arctan\left(\sqrt{|u|^2/|v|^2}\right) = \arctan\left(\sqrt{A_{\varepsilon_+}/A_{\varepsilon_-}}\right). \tag{3}$$

Here $u$ and $v$ are the respective particle and hole amplitudes of the Bogoliubov quasiparticles, which are related to the peak heights $A_{\varepsilon_\pm}$ at positive and negative peak energies $\varepsilon_\pm$ measured in tunnelling spectroscopy[34]. The results are shown in Fig. 4. For maximal particle–hole mixing ($|u|^2 = |v|^2$), the angle $\theta_B$ equals $\pi/4$. For Bogoliubov quasiparticles, this case is expected when their energy approaches the pairing energy $\varepsilon_\pm \approx \pm\Delta_{ind}$. In the experimental data, we indeed find a value of $\theta_B \approx \pi/4$ whenever $\bar{\varepsilon} \approx \varepsilon_{min}$ ($E_r \approx 0$; see Supplementary Note 2). This finding further supports the above conjecture that $\varepsilon_{min}$ can be interpreted as a proximity-induced superconducting pairing $\Delta_{ind}$ in the QD resonance level. When moving to larger in-gap state energies, $\theta_B$ either increases (for $E_r > 0$) or decreases (for $E_r < 0$). This trend is found consistently for all eigenmodes and qualitatively agrees well with the expectations for Bogoliubov excitation solutions of

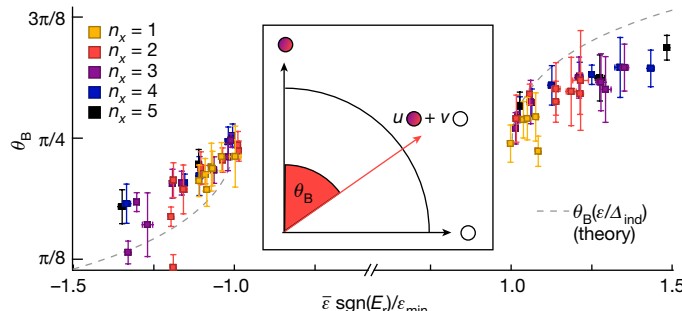

**Fig. 4 | Particle–hole mixture of the in-gap states.** Bogoliubov angle $\theta_B$ of the in-gap states with different mean energies $\bar{\varepsilon} = (\varepsilon_+ - \varepsilon_-)/2$ normalized to their minimal energies $\varepsilon_{\min}$. All error bars are standard deviations extracted from fitting the data; see Supplementary Note 2 for details. The dashed grey lines represent the expected relationship for Bogoliubov quasiparticles with an induced gap of $\Delta_{\text{ind}} = \varepsilon_{\min}$ as derived from the effective Hamiltonian in equation (2) (see Methods for details). Inset, Bogoliubov quasiparticles are coherent combinations of electrons (filled circle) and holes (empty circle). The Bogoliubov angle $\theta_B$ of a quasiparticle quantifies the amount of particle–hole mixing.

equation (2) (dashed grey lines in Fig. 4; see Methods and Extended Data Fig. 4 for details).

Our experimental observation of MSSs clearly challenges the idea that the appearance of sharp in-gap states in STM experiments on superconductors was a fingerprint of either a local magnetic moment[30,31] or unconventional superconducting pairing[32,33]. The sharp linewidth $\gamma$ of the in-gap states can be understood as a consequence of negligible scattering into the gapped bulk states, unlike in the metallic state at $E > \Delta_s$, in which the level obtains a broadening $\Gamma \gg \gamma$. The energy of the MSSs critically depends on the ratio of $\Gamma$ and $\Delta_s$. For typical localized levels residing on single-atomic impurities, this ratio is $\Gamma/\Delta_s \gg 1$ and, thus, the bound states are located at energies very close to the coherence peaks of the bath superconductor. Therefore, for atomic impurities coupled to superconducting hosts, these resonances are hardly detectable. However, the linewidths of the eigenmodes of the QDs studied here are of similar magnitude as the superconducting gap, which leads to the low-energy in-gap states depicted in Fig. 2c that are well split off from the coherence peaks.

The strongest coupling $\Gamma$ is observed for the narrowest investigated QD ($n_x = 1$ in Fig. 2d), resulting in a comparably large gap $\Delta_{\text{ind}}$ of up to 85% of $\Delta_s$ induced into the QD eigenmode. This suggests that the proximity effect originates from scattering of the surface state at the QDs walls, which is maximal for the narrowest QDs, as also speculated by recent works[16,17]. Because the coupling $\Gamma$ is controlled by the QD size, the induced gap $\Delta_{\text{ind}}$ is found to be tunable as well (see Fig. 2d). Moreover, as demonstrated in Fig. 4, the experimentally observed resonance peaks behave like Bogoliubov excitations, which are expected to carry an energy-dependent fractional charge[35] of $(|u|^2 - |v|^2)e$. This could potentially be directly examined by STM-based shot-noise measurements[36], opening avenues for studying quasiparticles with tunable fractional charge on the atomic scale.

We anticipate that the concept of impurity-supported proximity-induced Cooper pairing could be helpful in general to induce superconductivity into arbitrary surface states, potentially also combined with non-trivial topology. Among others, the latter presents a pathway for the creation of unconventional superconductivity and Majorana bound states[5,6,16,37]. Moreover, patterning the surface states of (111) noble-metal surfaces by precisely positioned scattering centres has evolved to one of the most promising platforms in the direction of artificial lattices. These have been shown to host Dirac fermions[38,39], flat bands[40–42], wavefunctions in fractal geometries[43] or topologically non-trivial states[42,44]. Eventually, our results facilitate studying the interaction of these exotic

phenomena with superconducting pairing in a simple and tunable platform. Notably, although electron–electron interactions inside the noble-metal QDs we study here are typically screened well by the charge carriers in the system's bulk, it would be interesting to extend this platform towards reduced screening, potentially enabling atomic-scale studies of the crossover from spin-degenerate to spinful QDs coupled to superconductors[45].

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

## Methods

### Experimental procedures

The experiments were performed in a commercially purchased SPECS STM system operated at $T = 4.5$ K, which is equipped with home-built ultrahigh-vacuum chambers for sample preparation[46]. STM images were obtained by regulating the tunnelling current $I_{stab}$ to a constant value with a feedback loop while applying a constant bias voltage $V_{stab}$ across the tunnelling junction. For measurements of differential tunnelling conductance ($dI/dV$) spectra, the tip was stabilized at bias voltage $V_{stab}$ and current $I_{stab}$ as individually noted in the figure captions. In a next step, the feedback loop was switched off and the bias voltage was swept from $-V_{stab}$ to $+V_{stab}$. The $dI/dV$ signal was measured using standard lock-in techniques with a small modulation voltage $V_{mod}$ (RMS) of frequency $f = 1,097.1$ Hz added to $V_{stab}$. The following measurement parameters have been used for the data presented in the main figures:

Fig. 1a: $V = 50$ mV, $I = 1$ nA, $V_{mod} = 5$ mV; Fig. 1c,d: $V = 5$ mV, $I = 1$ nA; Fig. 1e,f: $V_{stab} = -100$ mV, $I_{stab} = 2$ nA, $V_{mod} = 2$ mV; Fig. 2a,c: $V_{stab} = -15$ mV, $I_{stab} = 4$ nA, $V_{mod} = 50$ μV; Fig. 2b: $V_{stab} = -15$ mV, $I_{stab} = 4$ nA, $V_{mod} = 100$ μV.

$dI/dV$ line profiles were acquired recording several $dI/dV$ spectra along a one-dimensional line of lateral positions on the sample, respectively. Note that the tip was not stabilized again after each individual spectrum was acquired but the line profiles were measured in constant-height mode. This avoids artefacts stemming from a modulated stabilization height. At the chosen stabilization parameters, the contribution of multiple Andreev reflections and direct Cooper pair tunnelling to the superconducting tip can be neglected (see Supplementary Note 1). Throughout this work, we use Nb tips made from a mechanically cut and sharpened high-purity Nb wire. The tips were flashed in situ to about 1,500 K to remove residual contaminants or oxide layers. The use of superconducting tips increases the effective energy resolution of the experiment beyond the Fermi–Dirac limit[47] but requires careful interpretation of the acquired $dI/dV$ spectra. These are proportional to the convolution of the sample's LDOS, the superconducting tip DOS and the difference in the Fermi–Dirac distributions of the tip and sample. Notably, the latter can play a large role when measuring at $T = 4.5$ K. We measure a value of $\Delta_s = 1.35$ meV (Supplementary Note 1), which is similar to the gap of elemental Nb, $\Delta_{Nb} = 1.50$ meV (refs. 3,48), indicating a high interface quality between Nb and Ag. Details on the interpretation of SIS tunnelling spectra and on the determination of the tip's superconducting gap $\Delta_t$ can be found in the next section, as well as in Supplementary Note 1.

### Sample preparation

A Nb(110) single crystal was used as a substrate and cleaned by high-temperature flashes to $T \approx 2,000$ K. This preparation method yields the characteristic oxygen-reconstructed Nb surface observed in previous works[49], as can be seen in Extended Data Fig. 5a. Notably, a similar surface quality can be achieved by sputter-annealing cycles only, that is, without the need for challenging ultrahigh temperature flashes typically required for the preparation of clean c(1x1) Nb(110) surfaces[48]. Ag was deposited from an e-beam evaporator using a high-purity rod at a deposition rate of about 0.1 monolayers (ML) min$^{-1}$. In agreement with previous studies, evaporation of Ag at elevated temperatures leads to the formation of two pseudomorphic monolayers of Ag followed by Stranski–Krastanov growth of large Ag(111) islands (Extended Data Fig. 5c). To obtain preferably thin islands, we grew Ag islands in a three-step process, starting with the deposition of 2ML at $T \approx 600$ K, creating two closed wetting layers. In a second step, the temperature was reduced to $T \approx 400$ K to limit the lateral diffusion of Ag on the surface and to create more nucleation centres for the Stranski–Krastanov islands. Under these conditions, another 2ML of Ag were deposited, followed by three further ML grown at $T \approx 600$ K, again to guarantee a well-annealed surface of the topmost layers.

A topographic image of a Ag island grown on $NbO_x$/Nb(110) is shown in Extended Data Fig. 5a. This sample features a Ag coverage of only 15%, enabling the identification of the substrate's oxygen reconstruction (see refs. 48,49) and of the islands' apparent heights. Nearly all of such islands are found to have heights in the range 500–550 pm, consistent with a preferred double-layer growth. Low-energy $dI/dV$ spectroscopy measurements (Extended Data Fig. 5b) reveal clean SIS tunnelling on both the $NbO_x$/Nb(110) substrate and the Ag double-layer island: sharp peaks of high differential tunnelling conductance appear at bias voltages $eV = \pm(\Delta_t + \Delta_s)$, corresponding to quasiparticle tunnelling between the coherence peaks of the tip and sample, respectively. Also, weaker resonances are found at voltages $eV = \pm(\Delta_t - \Delta_s)$. These are typically attributed to thermally activated tunnelling between the partially occupied and unoccupied coherence peaks of the tip and sample[47]. From the positions of these peaks measured with different microtips, the tip and sample gaps can be unambiguously determined. Notably, there is no clear difference between the spectra measured on $NbO_x$/Nb(110) and on the Ag double layer, providing evidence that the interface quality between Nb and Ag is sufficient to open a full proximity gap in the Ag states.

As the Ag coverage is increased above 2ML, the Ag double layer is gradually closed and the formation of further large islands in the Stranski–Krastanov growth mode is observed (Extended Data Fig. 5c). For these samples, the closed double layer can be investigated in more detail. Characteristic defects of unknown chemical composition are found on the double layer, exhibiting a twofold symmetry (Extended Data Fig. 5d). This already suggests that the Ag film does not grow in a fcc(111) fashion for the first two layers. Instead, atomically resolved images of the double layer (Extended Data Fig. 5e) reveal a pseudomorphic growth of Ag on the bcc(110) surface of the underlying $NbO_x$/Nb(110). Previously, a similar growth mode has been reported for Ag layers on oxygen-reconstructed V(100) in ref. 50: the pseudomorphic nature of the growth despite a reconstructed substrate surface has been explained by the enhanced mobility of oxygen in vanadium at elevated temperatures, leading to a substitution of oxygen atoms by Ag atoms during the growth and to a clean interface. We speculate that a similar growth mode is taking place for the first double-layer Ag/Nb(110). In contrast to this, atomically resolved images on the thicker islands (Extended Data Fig. 5f) reveal the hexagonal lattice expected for the energetically favoured fcc(111) growth of Ag and a very low number of impurities (typically 1–2 per $100 \times 100$ nm$^2$; see Extended Data Fig. 6a). These results are in agreement with earlier reports of the growth mode of Ag on Nb(110) by low-energy electron diffraction[51] and STM[15,52].

On the islands, we find pronounced signatures of the well-known Shockley-type surface state of Ag(111), providing further evidence for a fcc(111) growth. In Extended Data Fig. 6b, an example of a constant-height $dI/dV$ map measured in the area of Extended Data Fig. 6a is shown, measured at an energy above the onset of the surface state and outside the superconducting gap. It features a clear periodic modulation in agreement with pronounced quasiparticle interference of the surface-state electrons[53,54]. A Fourier transformation of the map visualizes the circular Fermi surface of the surface state (inset of Extended Data Fig. 6b).

### Construction of QDs

As previously reported in ref. 55, approaching the tip to a Ag(111) surface can lead to two processes: (1) single Ag atoms can be reproducibly pulled out of the surface by attractive tip–sample interactions, leaving a vacancy behind in the Ag lattice, and (2) single Ag atoms are dropped from a Ag-coated tip, which was coated previously by dipping the tip into the Ag(111) surface.

An example of this process of adatom gathering is shown in Extended Data Fig. 7a–e. Once the Ag atoms are moved to a region without surface-state contributions—for example, inside a QD

structure (Extended Data Fig. 7f)−the d$I$/d$V$ spectra on top of Ag atoms (Extended Data Fig. 7g) show clean SIS tunnelling without signs of Yu−Shiba−Rusinov states[31]. This provides further evidence that the adatoms used for our QDs are indeed non-magnetic, as expected for Ag/Ag(111). Subsequently, after gathering a sufficient number of Ag atoms, the Ag QDs were constructed by lateral atom manipulation[56,57] at low tunnelling resistances of $R \approx 100$ kΩ. Because the Ag walls of the QDs have a finite transparency for the surface-state electrons, a second wall of Ag atoms is constructed around the central QD wall to screen the interior from surface-state modes located outside the structure.

## Modelling of QD eigenmodes

The wavefunctions of the eigenmodes of the QDs can be well modelled by hard-wall particle-in-a-box simulations to a first approximation, as it was done already in the pioneering work by Crommie et al.[11] in 1993. The eigenmodes of an infinitely high rectangular potential wall with dimensions $L_x$ and $L_y$ are the well-known analytical solutions:

$$\Psi(n_x, n_y) = \Psi_x(n_x) \times \Psi_y(n_y) \tag{4}$$

with

$$\Psi_j(n_j) = \sqrt{2/(L_j - \delta)} \times \begin{cases} \sin(\pi n_j/(L_j - \delta) \times j) \text{ for even } n_j \\ \cos(\pi n_j/(L_j - \delta) \times j) \text{ for odd } n_j \end{cases} \tag{5}$$

$j = x, y$ and the quantum numbers $n_x$ and $n_y$, corresponding to the number of antinodes of a certain eigenfunction. These correspond to eigenenergies of

$$E(n_x, n_y) = \frac{\hbar^2}{2m_{\text{eff}}}[(\pi n_x/(L_x - \delta))^2 + (\pi n_y/(L_y - \delta))^2] + E_0. \tag{6}$$

Note that the parameter $\delta$ is introduced to renormalize the effective dimensions of the QDs, because the distances seen by the scattered quasiparticles are not necessarily given by the distances of the adatoms of the walls. Here, $m_{\text{eff}} = 0.58 m_e$, the surface-state band edge $E_0 = -26.4$ meV and $\delta = -0.28$ nm are used, as motivated in Supplementary Note 2. The LDOS patterns presented in Figs. 1e and 2b have been calculated as a sum of the individual eigenfunctions with a finite Lorentzian broadening of $\Gamma = 3$ meV acting on their eigenenergies:

$$\text{LDOS}(E) = \sum_{n_x, n_y} \frac{|\Psi(n_x, n_y)|^2}{1 + (E - E(n_x, n_y))^2/\Gamma^2}. \tag{7}$$

## MSS model

We start by considering a system of a single spatially extended spin-degenerate level coupled to an s-wave superconducting three-dimensional bath, being a generalization of the model introduced in ref. 12:

$$\mathcal{H} = \sum_{\mathbf{k},\sigma} \epsilon_{\mathbf{k}} c^\dagger_{\mathbf{k},\sigma} c_{\mathbf{k},\sigma} + \sum_{\mathbf{k},\sigma} (\widetilde{V}(\mathbf{k}) c^\dagger_{\mathbf{k},\sigma} d_\sigma + \widetilde{V}(\mathbf{k})^* d^\dagger_\sigma c_{\mathbf{k},\sigma}) + \sum_\sigma E_i d^\dagger_\sigma d_\sigma \\ - \Delta_s \sum_{\mathbf{k}} (c^\dagger_{\mathbf{k},\uparrow} c^\dagger_{-\mathbf{k},\downarrow} + c_{-\mathbf{k},\downarrow} c_{\mathbf{k},\uparrow}). \tag{8}$$

The Hamiltonian given in equation (1) and in ref. 12 is a special case of equation (8) for a perfectly localized impurity level ($\widetilde{V}(\mathbf{k}) = V = $ constant). Here and in the following, we set $\hbar = 1$.

We aim to calculate the LDOS at the local level. For that, we use the Green's function equations of motion in energy space[58]

$$E G_{a_i, a_j^\dagger}(E) = \delta_{ij} + \langle\langle[a_i, \mathcal{H}]; a_j^\dagger\rangle\rangle, \tag{9}$$

in which $G_{a_i, a_j^\dagger}(E) = \langle\langle a_i; a_j^\dagger\rangle\rangle$ is the shorthand notation for the usual retarded Green's function[58], for which $a_i$ is one of the operators $d$, $c_{\mathbf{k}}$ or their adjoint. The LDOS at the local level is

$$\text{LDOS}(E) = -\frac{1}{\pi}\text{Im}[G_{d_\uparrow, d_\uparrow^\dagger}(\omega) + G_{d_\downarrow, d_\downarrow^\dagger}(\omega)], \tag{10}$$

in which $\omega = E + i\delta E$ and $\delta E$ is a small and positive real number approximating the experimentally observed energy broadening. We obtain the Green's function by solving the system of equations of motion in equation (9) for the Hamiltonian in equation (8) after linearizing the dispersion around $E_F$, that is, $\epsilon_{\mathbf{k}} = v_F(k - k_F)$, with $v_F$ and $k_F$ being the Fermi velocity and momentum, respectively:

$$G_{d_\sigma, d_\sigma^\dagger}(\omega) = \frac{\omega + E_r - \sum_{\mathbf{k}} |\widetilde{V}(\mathbf{k})|^2 \frac{(\omega - \epsilon_{\mathbf{k}})}{(\omega^2 - \epsilon_{\mathbf{k}}^2 - \Delta_s^2)}}{G_1 - G_2} \tag{11}$$

with

$$G_1 = \left(\omega + E_r - \sum_{\mathbf{k}} \frac{|\widetilde{V}(\mathbf{k})|^2(\omega - \epsilon_{\mathbf{k}})}{(\omega^2 - \epsilon_{\mathbf{k}}^2 - \Delta_s^2)}\right)\left(\omega - E_r - \sum_{\mathbf{k}} \frac{|\widetilde{V}(\mathbf{k})|^2(\omega + \epsilon_{\mathbf{k}})}{(\omega^2 - \epsilon_{\mathbf{k}}^2 - \Delta_s^2)}\right)$$

and

$$G_2 = \left(\sum_{\mathbf{k}} \frac{\Delta_s \widetilde{V}(\mathbf{k}) \widetilde{V}(-\mathbf{k})}{(\omega^2 - \epsilon_{\mathbf{k}}^2 - \Delta_s^2)}\right)\left(\sum_{\mathbf{k}} \frac{\Delta_s (\widetilde{V}(\mathbf{k}) \widetilde{V}(-\mathbf{k}))^*}{(\omega^2 - \epsilon_{\mathbf{k}}^2 - \Delta_s^2)}\right)$$

The impurity, which is described by its coupling to the substrate $V(\mathbf{r})$, has a localization length $L_{\text{imp}}$ corresponding to the size of the QD, and drops to zero for $|\mathbf{r}| \gg L_{\text{imp}}$. We can therefore reasonably set the corresponding Fourier transform $\widetilde{V}(\mathbf{k})$ constant for momenta $k = |\mathbf{k}|$ in the interval $[k_F - \beta/L_{\text{imp}}, k_F + \beta/L_{\text{imp}}]$, in which $\beta$ is on the order of one, whereas its concrete value depends on the spatial details of the impurity coupling $V(\mathbf{r})$. In the following order-of-magnitude approximation, we set $\beta = 1$. From equation (11), we find that the physics of a spatially extended impurity does not differ from that of a localized impurity if $\frac{1}{\omega^2 - \epsilon_{\mathbf{k}}^2 - \Delta_s^2} \ll 1$ at $k = k_F \pm 1/L_{\text{imp}}$. Combining the last two formulas, we find that an extended impurity can be considered as localized if $\omega$ is within a few $\Delta_s$ from $E_F$, which is the case for the experiment in the main text, and if $\frac{v_F}{\Delta_s} = \xi \gg L_{\text{imp}}$, in which $\xi = v_F/\Delta_s$ is the proximitized superconducting coherence length in the Ag islands. For Ag, the Fermi velocities range from 0.518 to $1.618 \times 10^6$ m s$^{-1}$ (ref. 59), resulting in $\xi = 253$ to 789 nm, which is considerably larger than the maximal extent of our QDs reaching $L_{\text{imp}} = 24$ nm. The QD level can therefore be treated as a localized impurity. In this limit, the Green's function can be written as

$$G_{d_\sigma, d_\sigma^\dagger}(\omega) = \frac{\omega + E_r + \frac{\Gamma\omega}{\sqrt{\Delta_s^2 - \omega^2}}}{\omega^2\left(1 + \frac{2\Gamma}{\sqrt{\Delta_s^2 - \omega^2}}\right) - E_r^2 - \Gamma^2}, \tag{12}$$

in which $\Gamma = \pi V^2 D$, with $D = k_F^2 W/(2\pi^2 v_F)$ being the density of states per spin species of the substrate above the critical temperature at $E_F$ and $W$ is its volume. The LDOS is given by

$$\text{LDOS}(E) = -\frac{2}{\pi}\text{Im}\left[\frac{\omega + E_r + \frac{\Gamma\omega}{\sqrt{\Delta_s^2 - \omega^2}}}{\omega^2\left(1 + \frac{2\Gamma}{\sqrt{\Delta_s^2 - \omega^2}}\right) - E_r^2 - \Gamma^2}\right]. \tag{13}$$

We note the emergence of in-gap states as found in ref. 12. The energy $\varepsilon_+$ of this in-gap state for a range of values $E_r$ and $\Gamma$ is plotted in Extended Data Figs. 2 and 3. Recently, an LDOS of a localized impurity including further magnetic scattering has been derived[60]. In contrast to a metallic

bath, in which the scattering results in a spectral broadening of the local level, the superconducting bath induces superconductivity by proximity to the local level. Hence, when $E_r$ lies within the gap of the superconductor, the state at $E_r$ splits into two particle–hole-symmetric ones around $E_F$. Notably, for energy scales $E_r$ sufficiently larger than $\Delta_s$, equation (13) reduces to a typical Lorentzian LDOS of width $\Gamma$ at position $E_r$, as observed in the experiment.

The obtained spin-degenerate single-level Hamiltonian with proximity-induced pairing (equation (2)) is equivalent to the Green's function approach above to the second order in the coupling constant $V \propto \sqrt{\Gamma}$, as we show in the following.

### Derivation of the effective Hamiltonian

In this section, we derive an effective low-energy model for the electronic level valid when the bare energy of the spin-degenerate electronic level is close to the Fermi energy and the coupling to the superconducting bulk is smaller than the superconducting gap. We find that the level obtains proximity pairing and a correction in its chemical potential.

The Hamiltonian of a spin-degenerate electronic level locally coupled to a Bardeen–Cooper–Schrieffer s-wave superconductor is given in equation (1), which we repeat here for convenience

$$\mathcal{H} = \sum_{\mathbf{k},\sigma} \epsilon_{\mathbf{k}} c^{\dagger}_{\mathbf{k},\sigma} c_{\mathbf{k},\sigma} + \sum_{\mathbf{k},\sigma} V(c^{\dagger}_{\mathbf{k},\sigma} d_{\sigma} + d^{\dagger}_{\sigma} c_{\mathbf{k},\sigma}) + \sum_{\sigma} E_r d^{\dagger}_{\sigma} d_{\sigma} \\ - \Delta_s \sum_{\mathbf{k}} (c^{\dagger}_{\mathbf{k},\uparrow} c^{\dagger}_{-\mathbf{k},\downarrow} + c_{-\mathbf{k},\downarrow} c_{\mathbf{k},\uparrow}), \tag{14}$$

in which $c_{\mathbf{k},\sigma}$ are the annihilation operators in the superconducting bulk with momentum $\mathbf{k}$ and spin $\sigma$, $d_{\sigma}$ the annihilation operator of the electronic level with spin $\sigma$, $\epsilon_{\mathbf{k}}$ the dispersion relation in the bulk, $E_r$ the electric potential of the electronic levels and $V$ quantifies the local coupling between the electronic levels and the superconducting bulk.

To derive the low-energy model, we use the Schrieffer–Wolff transformation

$$S = \sum_{\mathbf{k},\sigma} \mathrm{sgn}(\sigma) A_{\mathbf{k}} d_{\sigma} c_{\mathbf{k},-\sigma} + B_{\mathbf{k}} d_{\sigma} c^{\dagger}_{\mathbf{k},\sigma} - \mathrm{h.c.} \tag{15}$$

where h.c. is the Hermitian conjugate, with $\mathrm{sgn}(\uparrow) = 1$, $\mathrm{sgn}(\downarrow) = -1$, and

$$A_{\mathbf{k}} = \frac{-V\Delta_s}{\epsilon_{\mathbf{k}}\epsilon_{-\mathbf{k}} + \Delta_s^2 - E_r^2}, \tag{16}$$

$$B_{\mathbf{k}} = \frac{V(\epsilon_{-\mathbf{k}} - E_r)}{\epsilon_{\mathbf{k}}\epsilon_{-\mathbf{k}} + \Delta_s^2 - E_r^2}, \tag{17}$$

to obtain the effective Hamiltonian

$$\mathcal{H}' = e^S \mathcal{H} e^{-S} = \mathcal{H}'_D + \mathcal{H}'_{SC} + \mathcal{O}(V^3). \tag{18}$$

The physics inside the superconducting gap is contained in the effective Hamiltonian $\mathcal{H}'_D$, which is that of a spin-degenerate electronic level with proximity-induced superconductivity

$$\mathcal{H}'_D = \sum_{\sigma} (E_r + E_{\mathrm{shift}}) d^{\dagger}_{\sigma} d_{\sigma} - \Delta_{\mathrm{ind}} (d^{\dagger}_{\uparrow} d^{\dagger}_{\downarrow} + d_{\downarrow} d_{\uparrow}), \tag{19}$$

with the induced gap $\Delta_{\mathrm{ind}}$ and the shift $E_{\mathrm{shift}}$ in the chemical potential

$$\Delta_{\mathrm{ind}} = -\sum_{\mathbf{k}} V A_{\mathbf{k}}, \tag{20}$$

$$E_{\mathrm{shift}} = \sum_{\mathbf{k}} V B_{\mathbf{k}}. \tag{21}$$

We approximate equations (20) and (21) by linearizing the dispersion relation $\epsilon_{\mathbf{k}}$ close to the Fermi momentum $k_F$ by

$$\epsilon_{\mathbf{k}} = v_F(k - k_F), \tag{22}$$

in which $v_F$ is the Fermi velocity of the superconductor and we only consider momenta within the range $[k_F - \Lambda, k_F + \Lambda]$. For a three-dimensional host superconductor, we find

$$\Delta_{\mathrm{ind}} = 2V^2 D \arctan\left(\frac{\Lambda v_F}{\sqrt{\Delta_s^2 - E_r^2}}\right) \frac{\Delta_s}{\sqrt{\Delta_s^2 - E_r^2}} \lesssim V^2 D \pi \frac{\Delta_s}{\sqrt{\Delta_s^2 - E_r^2}} = \Gamma \frac{\Delta_s}{\sqrt{\Delta_s^2 - E_r^2}}, \tag{23}$$

$$E_{\mathrm{shift}} = -E_r \frac{\Delta_{\mathrm{ind}}}{\Delta_s}, \tag{24}$$

in which $\Gamma$ and $D$ are defined as in the main text. We infer that the effective Hamiltonian of the spin-degenerate electronic level close to $E_F$ obtains a proximity-induced superconducting pairing. From equation (19), we calculate the energy $\varepsilon$ of the level and the hole weight $|v|^2$ of the negative-energy eigenvalue to be

$$\varepsilon = \pm\sqrt{E_r^2(1 - \Delta_{\mathrm{ind}}/\Delta_s)^2 + \Delta_{\mathrm{ind}}^2}, \tag{25}$$

$$|v|^2 = \frac{1}{2} - \frac{E_r\left(1 - \frac{\Delta_{\mathrm{ind}}}{\Delta_s}\right)}{2\varepsilon} = \frac{1}{2} - \frac{\sqrt{\varepsilon^2 - \Delta_{\mathrm{ind}}^2}}{2\varepsilon}, \tag{26}$$

in which we have neglected orders of $\Delta_{\mathrm{ind}}^3$ and higher in the last step. If the spin-degenerate electronic level originally lies at the Fermi energy, that is, $E_r = 0$, its effective Hamiltonian only contains induced superconductivity and its Bogoliubov quasiparticles have 50% particle and 50% hole content. Moreover, the resonances are located at $\pm\varepsilon_{\mathrm{min}} = \pm\Delta_{\mathrm{ind}}$ for $E_r = 0$. Thus, the proximity-induced pairing strength can be readily inferred from measuring the value of $\varepsilon_{\mathrm{min}}$.

Using $|u|^2 = 1 - |v|^2$ for the particle weight, the Bogoliubov angle, which conveniently measures the amount of particle–hole mixing, takes the form

$$\theta_B(\varepsilon) = \arctan(\sqrt{|u|^2/|v|^2}) = \arctan\left(\sqrt{\frac{1 + \sqrt{\varepsilon^2 - \Delta_{\mathrm{ind}}^2}/\varepsilon}{1 - \sqrt{\varepsilon^2 - \Delta_{\mathrm{ind}}^2}/\varepsilon}}\right). \tag{27}$$

Notably, equation (27) is independent of $\Delta_{\mathrm{ind}}$ if the energies $\varepsilon$ are normalized by $\Delta_{\mathrm{ind}}$. This is the reason why equation (27) is used to plot the theoretical curve in Fig. 4. For the Bogoliubov angle of the MSSs based on the LDOS given in equation (13), the energy-dependent $\theta_B(\varepsilon)$ varies with $\Gamma$ and is thus different for each eigenmode. In Extended Data Fig. 4, we compare the Bogoliubov angle for a single superconducting level (equation (27)) with the expected Bogoliubov angle of MSSs using the expression for the LDOS calculated in equation (13). In the low-energy limit, both theories agree well, verifying that the anticrossing of the MSSs is evidence for superconducting pairing in the spin-degenerate level. For higher energies, the MSSs approach the coherence peak of the bulk gap and their asymmetries decrease again and finally converge to zero (equivalent to $\theta_B$ approaching $\pi/4$ at $\Delta_s$, marked by the dashed blue lines in Extended Data Fig. 4). This leads to an even better agreement with the experimental data and demonstrates that the observed resonances indeed behave like MSSs.

## Data availability

The raw data presented in the main figures of this work as well as the corresponding source data are available on Zenodo (https://doi.org/10.5281/zenodo.7971149). Further data supporting the findings of this study are available from the authors on reasonable request.

59. Mitchell, J. W. & Goodrich, R. G. Fermi velocities in silver: surface Landau-level resonances. *Phys. Rev. B* **32**, 4969–4976 (1985).
60. Villas, A. et al. Interplay between Yu-Shiba-Rusinov states and multiple Andreev reflections. *Phys. Rev. B* **101**, 235445 (2020).

## Code availability

The analysis codes that support the findings of the study are available from the corresponding authors on reasonable request.

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

**Acknowledgements** L.S., I.I., J.N.-S., J.W. and R.W. gratefully acknowledge funding by the Cluster of Excellence 'Advanced Imaging of Matter' (EXC 2056 – project ID 390715994) of the Deutsche Forschungsgemeinschaft (DFG). K.T.T., J.W. and R.W. acknowledge support by the DFG – SFB-925 – project 170620586. T.P. acknowledges support by the DFG (project no. 420120155). R.W. acknowledges funding of the European Union through the ERC Advanced Grant ADMIRE (project no. 786020). We thank P. Beck, H. Kim, R. Lo Conte, T. Wehling, M. Potthoff and A. Weismann for helpful discussions.

**Author contributions** L.S. and J.W. conceived the experiments. L.S. and K.T.T. performed the measurements and analysed the experimental data. L.S. simulated the QD eigenmodes using the hard-wall model. I.I. derived the resonance scattering model and J.N.-S. performed the numerical tight-binding simulations, both under the supervision of T.P. T.P. derived the low-energy model. L.S. prepared the figures. L.S. and J.W. wrote the manuscript. All authors contributed to the discussions and to correcting the manuscript.

**Funding** Open access funding provided by Universität Hamburg.

**Competing interests** The authors declare no competing interests.

### Additional information

**Correspondence and requests for materials** should be addressed to Lucas Schneider.

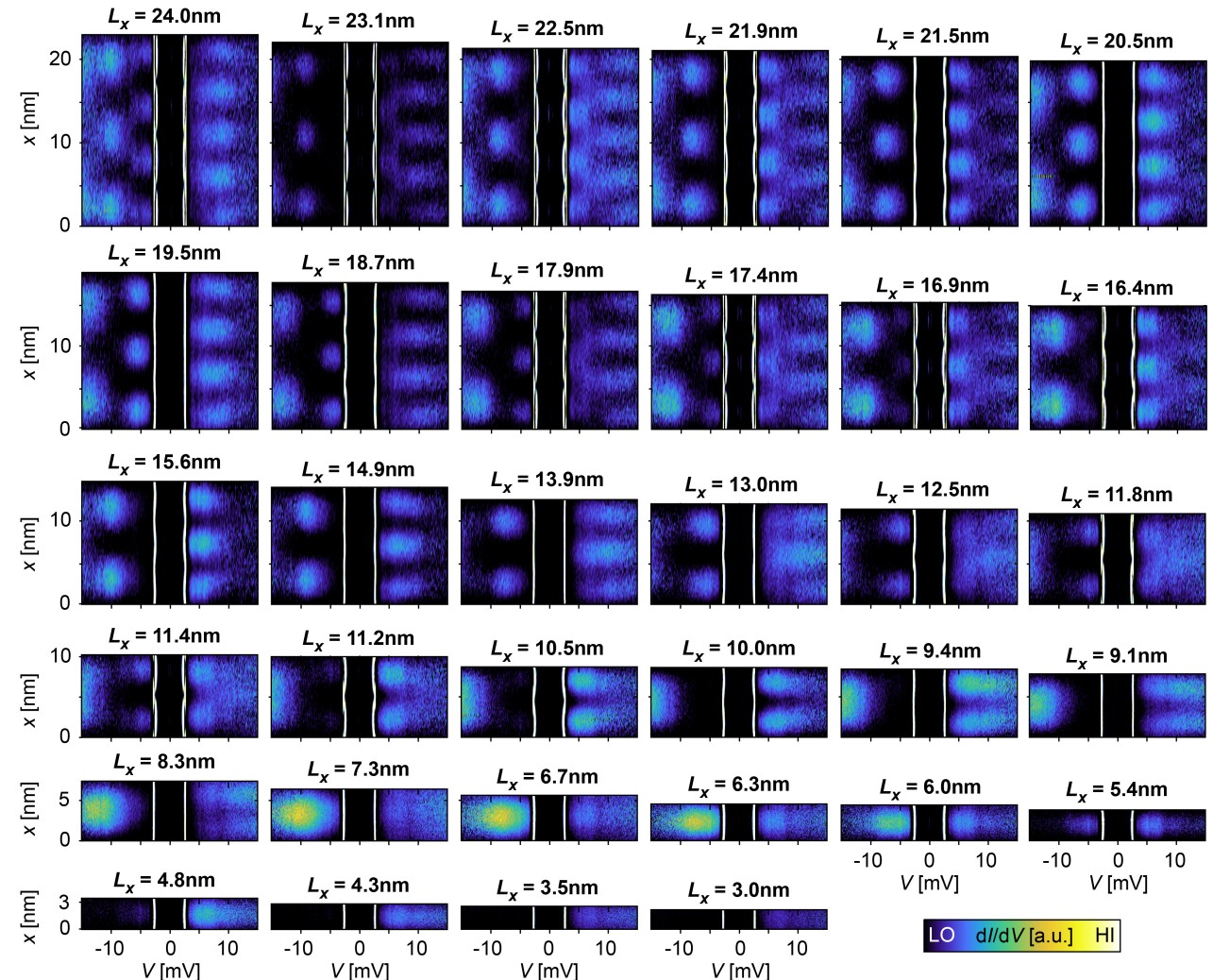

**Extended Data Fig. 1 | Complete set of d*I*/d*V* line profiles measured on 34 different QDs.** d*I*/d*V* line profiles measured along the central vertical axis of 34 different QDs with lengths $L_x$ ranging from 3.0 nm up to 24.0 nm. All line profiles have been measured at constant tip height. Parameters: $V_{stab}$ = −15 mV, $I_{stab}$ = 4 nA, $V_{mod}$ = 50 µV.

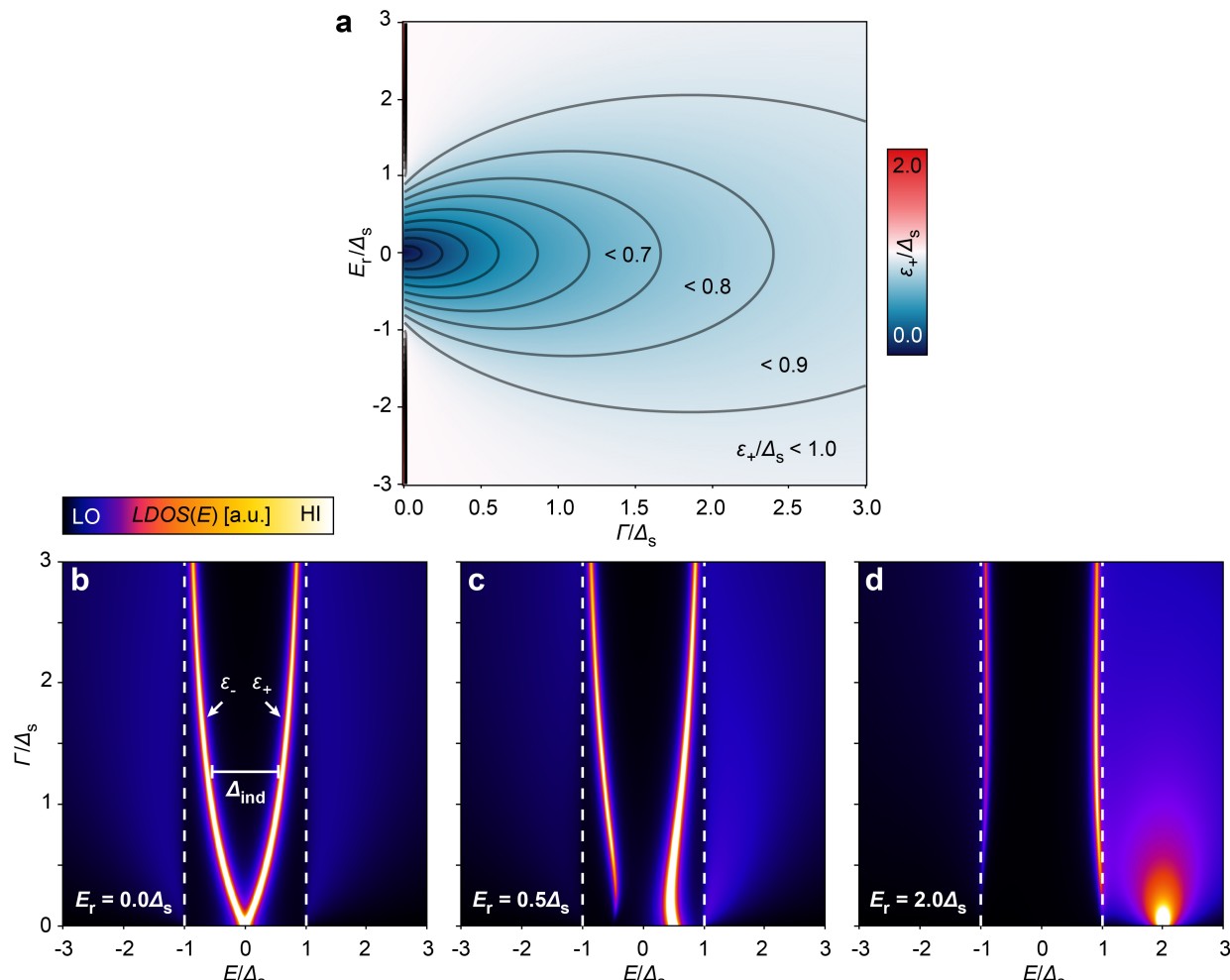

**Extended Data Fig. 2 | MSS energy versus localized level energy $E_r$ and coupling strength $\Gamma$ to the superconducting host. a**, Energy of the lowest-energy excitation of the Hamiltonian in equation (1). For all $\Gamma \neq 0$, an in-gap solution with $\varepsilon_+/\Delta_s < 1$ is found. **b**–**d**, Energy-dependent local electron density of states LDOS($E$) of a spin-degenerate localized level at energy $E_r/\Delta_s = 0.0$ (panel **b**), $E_r/\Delta_s = 0.5$ (panel **c**) and $E_r/\Delta_s = 2.0$ (panel **d**) coupled to a superconducting bath with the order parameter $\Delta_s$ by a coupling strength $\Gamma \propto V^2$ (see Methods). An energetic broadening of $\delta E = 0.03\Delta_s$ has been added in all panels (see Methods).

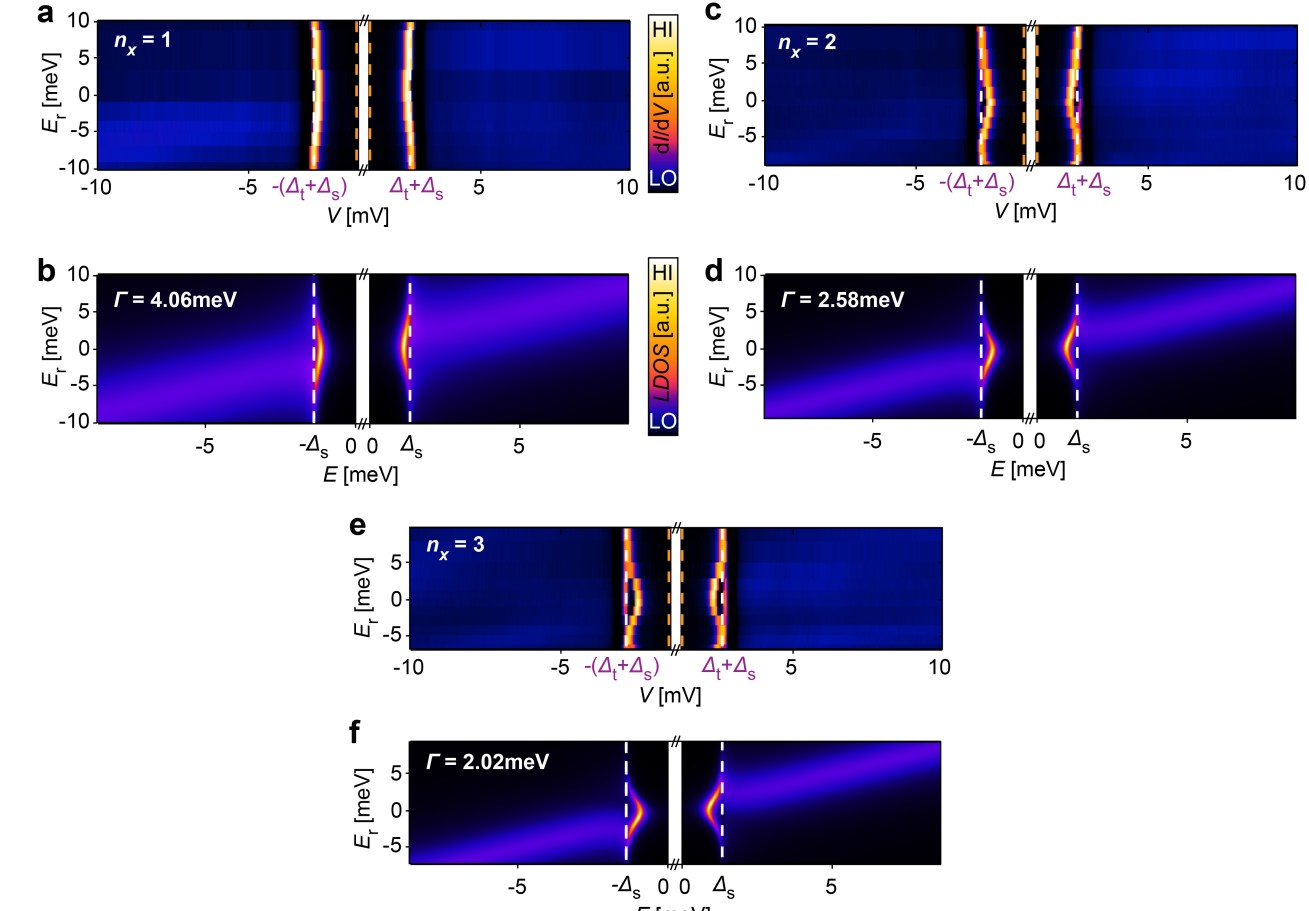

**Extended Data Fig. 3 | Dependence of MSS energy on the localized level energy $E_r$ for individual QD eigenmodes. a**, Evolution of averaged d$I$/d$V$ spectra from d$I$/d$V$ line profiles measured along the central vertical axis of different QDs (same data as Fig. 2c) as a function of the localized level energy $E_r$ of the $n_x = 1$ resonance. The value of $E_r$ has been extrapolated from inserting the known QD length into the fit function for $E_r(L_x)$ obtained in Supplementary Note 2. **b**, Energy-dependent local electron density of states LDOS($E$) of a single localized level at energy $E_r$ coupled to a superconducting bath with the parameter $\Delta_s = 1.35$ meV. The coupling strength $\Gamma$ is set to 4.06 meV, motivated by the average experimental width of the $n_x = 1$ resonances taken from Fig. 2d. **c**,**d**, Same as panels **a** and **b** but for the $n_x = 2$ resonances and $\Gamma = 2.58$ meV. **e**,**f**, Same as panels **a** and **b** but for the $n_x = 3$ resonances and $\Gamma = 2.02$ meV. An energetic broadening of $\delta E = 0.08$ meV corresponding to the experimental energy resolution has been added in all theoretical panels (see Methods).

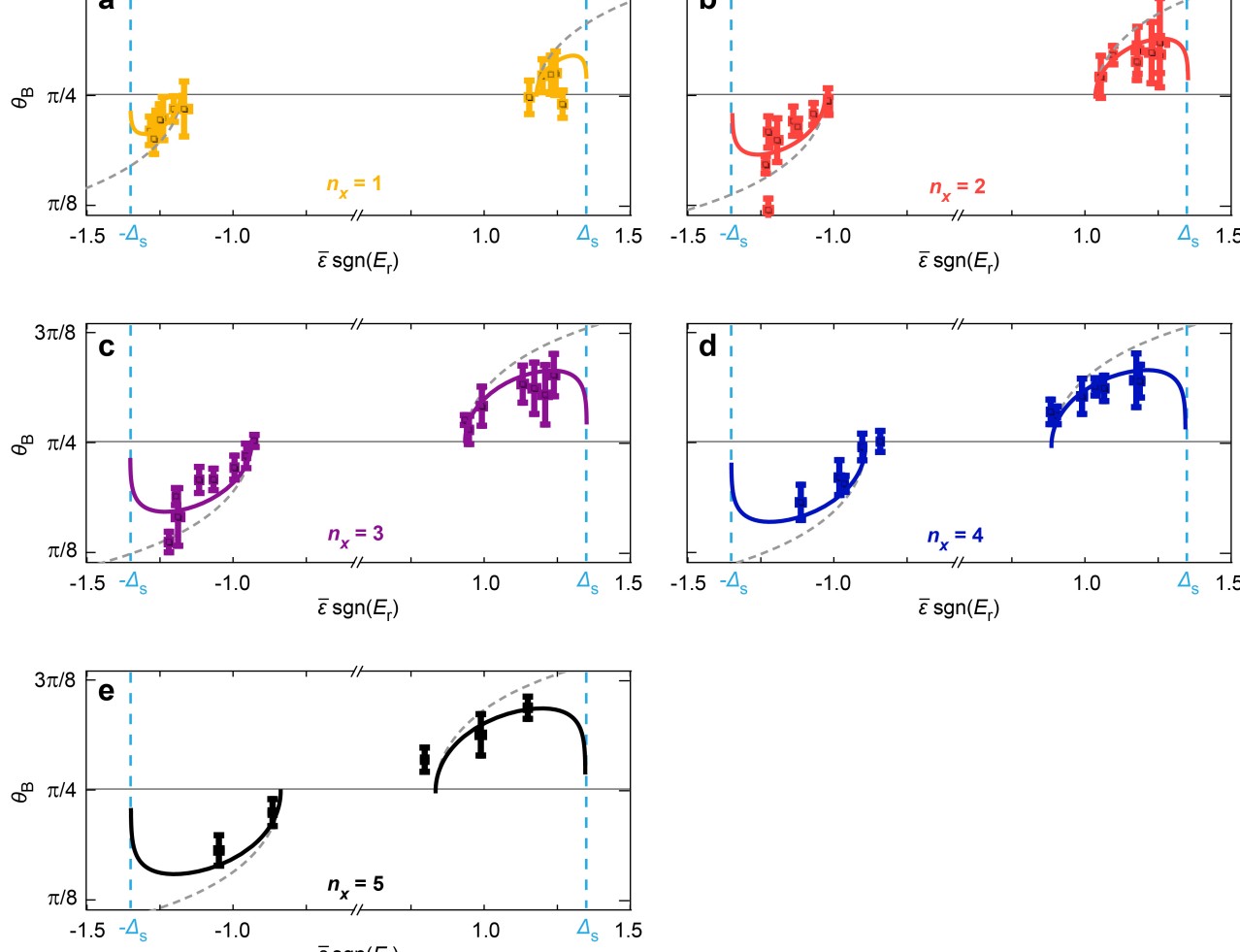

**Extended Data Fig. 4 | Bogoliubov angle of individual eigenmodes compared with the MSS model. a**, Bogoliubov angle $\theta_B$ of the in-gap states of the $n_x = 1$ eigenmode with different mean energies $\bar{\varepsilon} = (\varepsilon_+ - \varepsilon_-)/2$. All error bars are standard deviations extracted from fitting the data; see Supplementary Note 2 for details. The coloured lines represent the energy-dependent Bogoliubov angles of MSSs computed numerically from the LDOS in equation (13) in Methods. Here we use $\Gamma = 4.55$ meV as extracted on average for all $n_x = 1$ eigenmodes

(see Supplementary Note 2) and the experimental value $\Delta_s = 1.35$ meV. The dashed grey lines represent the expected relationship for Bogoliubov quasiparticles (equation (27)) with an induced gap of $\Delta_{ind} = \varepsilon_{min}$ set to the value given by the induced gap of the MSS model using the values for $\Gamma$ and $\Delta_s$ above. **b**, The same for the $n_x = 2$ eigenmodes and $\Gamma = 2.82$ meV. **c**, $n_x = 3$ eigenmodes, $\Gamma = 2.20$ meV. **d**, $n_x = 4$ eigenmodes, $\Gamma = 1.96$ meV. **e**, $n_x = 5$ eigenmodes, $\Gamma = 1.73$ meV. The sample gap $\Delta_s$ is marked by the light blue dashed lines in all panels.

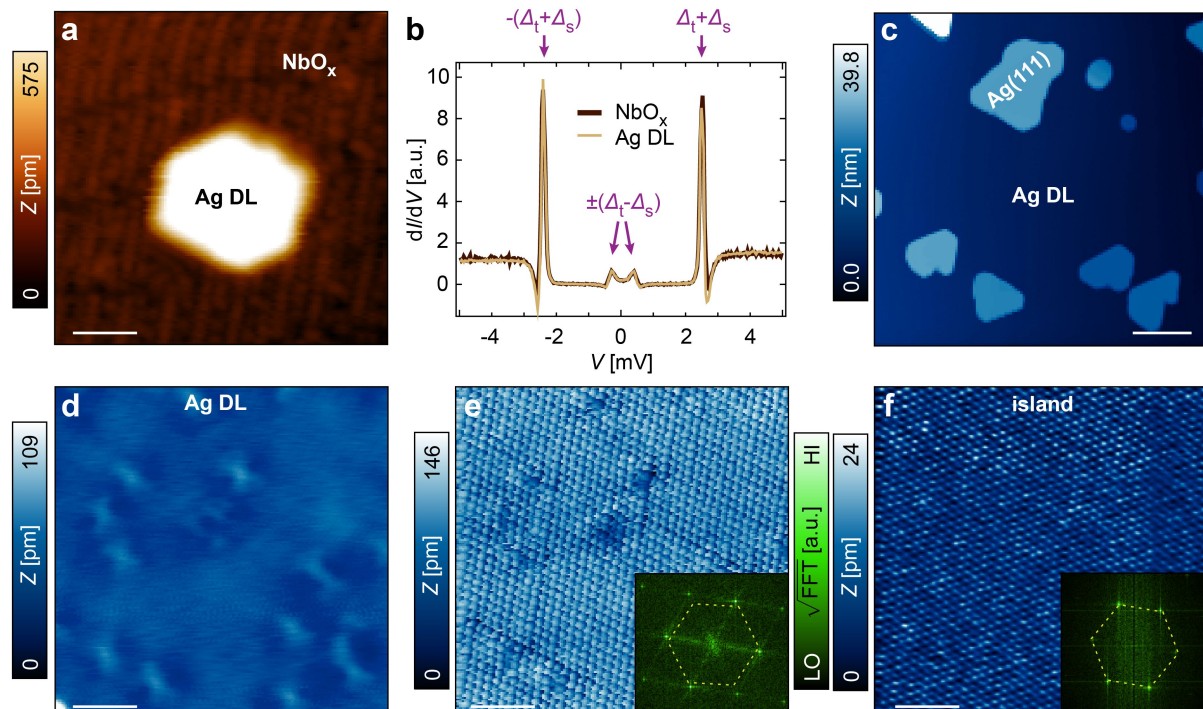

**Extended Data Fig. 5 | Growth of Ag on Nb(110). a**, Constant-current STM image of a thin Ag island grown on oxygen-reconstructed Nb(110). The apparent height of the island equals 540 pm, indicating that Ag grows in double layers (DL). The white bar corresponds to 2 nm. **b**, d$I$/d$V$ spectra measured on the Ag DL and on the oxidized Nb(110) substrate. The sharp peaks at bias voltages corresponding to $\pm(\Delta_t + \Delta_s)$ and $\pm(\Delta_t - \Delta_s)$ are marked by purple arrows. **c**, Large-scale constant-current image of a sample with nominal Ag coverage of 8ML. The DL is found to cover the entire Nb surface and further thicker Ag(111) islands are formed. The white bar corresponds to 500 nm. **d**, Zoom-in on the DL surface quality, exhibiting atomically flat areas of Ag and several twofold symmetric defects of unknown origin. The white bar corresponds to 2 nm.

**e**, Atomically resolved constant-current image of the same area shown in panel **d**. Inset, Fourier transform of the atomic-resolution image, showing Bragg spots incompatible with a hexagonal lattice (a perfect hexagon is overlaid as a yellow dashed line) but with a pseudomorphic growth on the bcc(110) surface of clean Nb. The white bar corresponds to 2 nm. **f**, Atomically resolved constant-current image of a thick Ag island. The white bar corresponds to 2 nm. Inset, Fourier transform of the image, showing Bragg spots in good agreement with a hexagonal fcc(111) growth. Parameters: $V$ = 5 mV, $I$ = 1 nA for panel **a**; $V_{stab}$ = 5 mV, $I_{stab}$ = 1 nA, $V_{mod}$ = 50 μV for panel **b**; $V$ = 1,000 mV, $I$ = 0.1 nA for panel **c**; $V$ = 2.5 mV, $I$ = 10 nA for panel **d**; $V$ = 2.5 mV, $I$ = 80 nA for panel **b**; $V$ = 100 mV, $I$ = 1 nA for panel **f**.

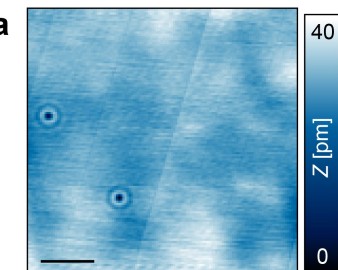

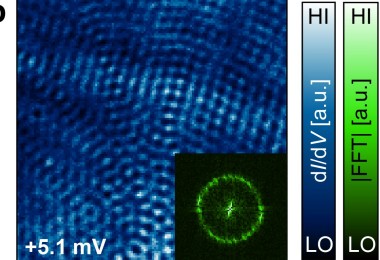

**Extended Data Fig. 6 | Low-energy electronic structure of Ag(111)/Nb(110). a**, Large-scale constant-current STM image of a 12 nm thin Ag island. The black bar corresponds to 20 nm. **b**, d$I$/d$V$ map at above-gap energy acquired with constant tip height and on the same region shown in panel **a**. The insets show the Fourier transform (FFT) of the d$I$/d$V$ map. Parameters: $V = -100$ mV, $I = 1$ nA for panel **a**; $V_{stab} = 10$ mV, $I_{stab} = 2$ nA, $V_{mod} = 100$ μV for panel **b**.

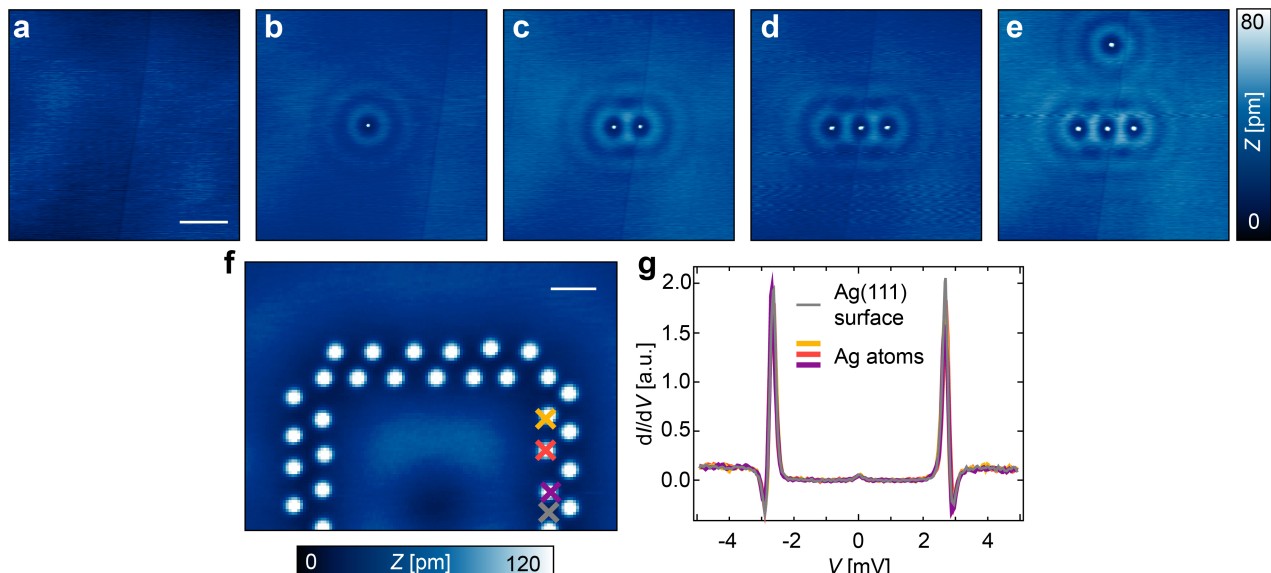

**Extended Data Fig. 7 | Single Ag atoms on Ag(111)/Nb(110). a**, Constant-current STM image of a clean Ag area. The white bar corresponds to 10 nm. **b–e**, The same area after approaching the tip by 600 pm towards the surface at different positions. A single adsorbate of similar apparent height is found after each approach with the tip, consistent with the finding of ref. 55. Therefore, we identify these adsorbates as single Ag atoms. **f**, Constant-current STM image of a QD's wall consisting of Ag atoms manipulated to form a suitable shape of the wall. The white bar corresponds to 2 nm. **g**, d$I$/d$V$ spectra measured on three of the Ag atoms marked in panel **f** (yellow, purple and red) as well as on the surface between the atoms (grey). Parameters: $V = 15$ mV, $I = 1$ nA for panels **a**–**e**; $V = 5$ mV, $I = 1$ nA for panel **f**; $V_{stab} = -5$ mV, $I_{stab} = 1$ nA, $V_{mod} = 50$ μV for panel **g**.