## [Peer Review File · Nature]

Manuscript Title: Proximity superconductivity in atom-by-atom crafted quantum dots

Reviewer Comments & Author Rebuttals

Reviewer Reports on the Initial Version:

Referees' comments:

Referee #1 (Remarks to the Author):

In the past decade or so, there have been a large amount of works dedicated to the study of bound states in superconductors. Among them, one finds Andreev bound states which have been extensively studied in mesoscopic hybrid systems (i.e. in various quantum dots, break junctions, carbon nanotubes and semi-conducting wires connected to superconducting leads), Yu-Shiba-Rusinov bound states induced by magnetic impurities in superconductors or the controversial Majorana zero modes. In this paper, Schneider et al. report in-gap bound states in the gap of a proximitized superconductor induced by non-interacting resonant levels predicted fifty years ago by Machida and Shibata. Though the theoretical study of such simplest bound states are now textbooks (a whole section is devoted to it in the classical review by Balatsky, Vekhter, and Zhu, Rev. Mod. Phys. 78, 373, (2006)), they have not been observed in their essence in such a control manner in a STS experiment, to the best of my knowledge. The main reason is that a local non-magnetic impurity (adatom) has an hybridization much larger than the superconducting gap; this entails that the associated bound state is typically lying right at the gap edge and cannot be spectroscopically identified.

In order to circumvent this difficulty, the authors build artificial quantum dots (or quantum billiards) made of Ag atoms adsorbed on a thin Ag(111) double layer grown on superconducting Nb(110). Because of the intermediate Ag thin layers, it is found that the modes of these billiards hybridized with the superconducting Nb substrate with an hybridization of the order of the proximitized gap. This gives rise to pairs of intra-gap bound states, which are detached from the gap-edge and thus observable! Similar conclusions are also reached for natural confined areas in the Ag(111) thin layer in Sec. S5. which reinforces this interpretation.

Let me first state that the surface and whole sample preparation and characterization is done meticulously according to Fig. 1 and the additional information given in the SI. The whole set of data concerning the modes of the quantum dot and the associated in-gap bound states shown in this paper is very clear, neat, consistent and convincing. The whole body of work behind this manuscript and the quality of the data is therefore impressive! I have no doubt that the results presented in this manuscript would be of high interest for the STM community.

My main concern is related to its interpretation and degree of novelty.

The authors build their argumentation on the fact that the Machida-Shibata bound state has not been detected. However, giving an unusual name to a bound state does not make it new! Though it is true that Machida and Shibata were the first to perform a theoretical calculation for a resonant level in a superconductor, bound states related to (interacting) resonant levels have been studied both theoretically and experimentally in numerous mesoscopic systems.

For example there have been numerous papers studying various quantum dots systems in proximity of superconducting leads. I am quoting only these two: JD Pillet et al., Nature Physics 6, 965 (2010); T. Dirks, Nature Physics 7, 386 (2011) but many have followed. The main difference compared to the present manuscript is the presence of local Coulomb interactions (i.e. a finite U Anderson model is used to interpret the data) and the bound states studied in the aforementioned papers carry the more conventional name of Andreev bound states. In this sense, the Machida-Shibata bound states can be regarded as a particular case of Andreev bound states. This is not

only a question of semantics. It is legitimate to ask how the present work compare to previous studies about Andreev bound states in various quantum dots ? In particular, the authors should state what are the new physical features (and there are some) the present study bring comparatively.

More specific questions and comments:

i) The authors measure a proximity -induced superconducting gap of order 1.35meV in the thin-Ag islands. How homogeneous is the gap ? Along the same line, is superconductivity in AgDL in the ballistic regime or diffusive one or somewhat intermediate?

ii) Data were taken at 4.5K with a superconducting tip to increase energy resolution. Even with such a tip, I would have expected some substantial thermal broadening of the in-gap states. However the in-gap states in Fig 2 look very sharp. Is the measured width consistent with thermal broadening ? Details would be welcome.

iii) The authors write `` The observation of in-gap states is a surprising result, since particle-hole-symmetric states inside the gap of an s-wave superconductor are commonly believed to only appear for magnetic impurities''

This is the author's opinion. As I wrote already, there is whole section devoted to it in the classical review by Balatsky, Vekhter, and Zhu, Rev. Mod. Phys. 78, 373, (2006)) and Andreev bound states have been largely studied in other mesoscopic systems.

iv) In Fig. 4, the data deviate significantly from the theoretical curve for $|\epsilon/\epsilon_{\min}| > 1.2$. Do the authors understand why ? Is the theoretical model oversimple or the assumption $\epsilon_{\min} \sim \Delta_{\text{ind}}$ too crude ?

v) It is not a priori obvious that the simple Machida-Shibata (MS) model used in the analysis would apply to describe the bound states associated to delocalized modes of the billiard. However, the complementary study given in Sec. S6 seems to confirm that the modes of the billiard can be indeed proximitized. Can the authors show the consistency of their chosen parameters with the experimental ones (by comparing dimensionless ratios) ?

vi) Furthermore, I do not see how I can reconcile the nice study of Sec. VI with the simple MS model. Can the authors establish how the parameters of the MS model be related to the model presented in S6 ?

Referee #2 (Remarks to the Author):

Summary: Schneider and co-authors present a comprehensive study of confined states in quantum corrals and their interaction through the proximity effect with a superconducting order parameter. The manuscript is well presented and accessible, the conclusions are supported by the experimental data and further backed by extensive additional analysis and supporting material. I recommend publication of the article as-is, with no need for further revision. My assessment is based on the following considerations:

A - Key Results

To my mind, the key results of the work are:

- 1.- The experimental demonstration of sub-gap states from a spin-degenerate impurity level coupled to a superconductor
- 2.- Their explanation through a theoretical model
- 3.- Establishing surface-bulk scattering as the mediator for the proximity effect in surface states.

B - Originality and Significance

As the authors note, the proximity effect is a key ingredient for designer quantum materials. The most notable example in the recent past, at least to my mind, would be the discovery of Majorana

bound states in a variety of proximitised nanostructures. The authors give several further examples in the manuscript. From this point of view, a systematic study of the proximity effect from minimal ingredients is desirable and may inspire new generations of designer quantum states. In that vein, quantum corrals are perhaps the dominant building block to study artificial lattices in the STM, but have not yet been paired with superconductivity. The authors show that the confined levels do indeed produce localised states which are affected the proximity effect and could serve as building blocks for exotic states of matter. Sub-gap states in superconductors have been extensively studied by local probes. The field focuses exclusively on the Yu-Shiba-Rusinov states to the point that sub-gap states are seen as direct evidence of spinful impurity levels. The present work challenges this view rather directly and introduces a new kind of sub-gap state with its own physics. While the results of the paper are immediately relevant in the context of local probes and artificial lattices, I believe that the fundamental physics will be interesting to a much wider audience.

C - Data and Methodology

The experiment is well designed and executed. By adjusting the size of the corral, the authors are able to tune the energy of the resonance states in their system and collect data over a wide parameter space. Data is of excellent quality, clearly presented, thoughtfully analysed, and supports the conclusions.

D - Appropriate use of statistics and treatment of uncertainties

It is not quite clear to me whether the error bars in figures 2d and 4 conform to the editorial policy of being 'defined in the corresponding figure legends', please cross-check this.

E - Conclusions

The conclusions follow from the analysis presented in the paper. They are supported by the data and further backed by extensive additional analysis and derivations in the supporting material.

F - Improvements

As a minor note, the figures, the axis labels in particular, appear heavily pixelated in the current format when the view is stretched to the width of the screen. I assume that this is because the figures are prepared according to the guidelines for publication in single- or double-column format, but please make sure they have sufficient resolution to be easily legible in the publication format.

G - References

The paper contains an extensive list of references touching on all relevant topics.

H - Clarity and context

I find the paper well-presented, clear, and accessible.

I - Inflammatory material

I find no cause for concern in this article.

Referee #3 (Remarks to the Author):

Schneider et al investigate superconductivity in quantum dots crafted on a proximitized superconducting surface. They observe states within the superconducting energy gap that shift depending on the size of the quantum corral. They interpret these states as Machida-Shibata (MS) states, which have been predicted decades ago, but were assumed to lie extremely close to the gap edge when single-atom scatterers are considered. Hence, the observation and tunability of these states is surprising and interesting. This being said, I am not yet convinced about the correct interpretation of the origin of these states. In my opinion there is not sufficient evidence from the data that the states are not conventional Andreev bound states which can be tuned by the

coupling energy, similar to what is done in conventional transport experiments using quantum dots.

A large part of the arguments for the assignment to MS states is based on the correlation of the energy of the MS states and the coupling energy of the quantum-well states to the bulk substrate. However, the analysis of the coupling energy does not seem to be appropriate in this context. The relevant coupling energy should be the one of the state crossing the Fermi level. Contrary to this, the authors deduce the coupling strength from a bunch of measurements including those of higher-lying states in their analysis.

Looking at Figure 2c, I am also not convinced by the position of the white dashed lines. The data in the background seems to be much flatter than the indicated L_x^{-2} behavior. Additionally, the lines do not seem to cross the gap at the same time as the minimum energy of the subgap state is reached. Possibly, this is just not clear from the data representation, and can be resolved by plotting the individual spectra in a stack (at least as Extended Data Figure) and showing the peak heights and shifts more explicitly.

Another problem is that the width of the quantum well states seems to be much broader than shown in Figure 2d. It occurs to me as if the peaks are at least 5mV in width independent of the quantum number n_x . Again, a more detailed presentation of the data may help. I am aware that Supplementary Note 4 addresses the determination of the width, but it also only shows the data of one corral in Figure S5a-c, which is even an example, where there is no resonance crossing the Fermi level and the width of $n=3$ is broader than $n=2$, contrary to the extracted value of the dashed line in Fig. S5d. The authors argue that this is due to the finite lifetime, which is correct but this problem could be avoided by only considering the quantum well states at the Fermi level.

Overall, it is not clearly specified from which corrals the data in Figure S5 d is derived and which states are taken into account (see comment above). The data scatters quite significantly and it is not clear why the authors draw the constant dashed lines. There could possibly be a dependence of Γ on E instead of being constant. As Γ is one of the most important parameters, this analysis should be substantiated.

The comparison of experiment and theory is not laid out clearly. Instead of plotting three graphs of different coupling strengths a direct comparison to experiment would better be compiled in one graph showing the MS state as a function of coupling strength.

Additionally, in experiment the minimal energies are plotted as a function of coupling strength (Fig. 2d). An additional plot of the MS states as a function of energy of the QWS would be helpful for direct evidence of the avoided crossing and direct comparison to Fig. 3. Such representation may be better suited to show the avoided crossing.

Author Rebuttals to Initial Comments:

Yellow marked text = here we added or changed something in the manuscript/supplement
The corresponding changes appear as **red text** in the manuscript/supplement

Referee #1 (Remarks to the Author):

In the past decade or so, there have been a large amount of works dedicated to the study of bound states in superconductors. Among them, one finds Andreev bound states which have been extensively studied in mesoscopic hybrid systems (i.e. in various quantum dots, break junctions, carbon nanotubes and semi-conducting wires connected to superconducting leads), Yu-Shiba-Rusinov bound states induced by magnetic impurities in superconductors or the controversial Majorana zero modes. In this paper, Schneider et al. report in-gap bound states in the gap of a proximitized superconductor induced by non-interacting resonant levels predicted fifty years ago by Machida and Shibata. Though the theoretical study of such simplest bound states are now textbooks (a whole section is devoted to it in the classical review by Balatsky, Vekhter, and Zhu, Rev. Mod. Phys. 78, 373, (2006)), they have not been observed in their essence in such a control manner in a STS experiment, to the best of my knowledge. The main reason is that a local non-magnetic impurity (adatom) has an hybridization much larger than the superconducting gap; this entails that the associated bound state is typically lying right at the gap edge and cannot be spectroscopically identified.

In order to circumvent this difficulty, the authors build artificial quantum dots (or quantum billiards) made of Ag atoms adsorbed on a thin Ag(111) double layer grown on superconducting Nb(110). Because of the intermediate Ag thin layers, it is found that the modes of these billiards hybridized with the superconducting Nb substrate with an hybridization of the order of the proximitized gap. This gives rise to pairs of intra-gap bound states, which are detached from the gap-edge and thus observable! Similar conclusions are also reached for natural confined areas in the Ag(111) thin layer in Sec. S5. which reinforces this interpretation.

Let me first state that the surface and whole sample preparation and characterization is done meticulously according to Fig. 1 and the additional information given in the SI. The whole set of data concerning the modes of the modes of the quantum dot and the associated in-gap bound states shown in this paper is very clear, neat, consistent and convincing. The whole body of work behind this manuscript and the quality of the data is therefore impressive! I have no doubt that the results presented in this manuscript would be of high interest for the STM community.

We thank the Reviewer for appreciating our work's quality and the novelty of our observations. As a minor remark, we would like to point out that the section in the Rev. Mod. Phys. review focusing on the work of Machida and Shibata is not even half a page long and basically concludes that the sub-gap resonances are in fact expected to be energetically positioned so close to the gap edge that they are irrelevant. However, we prove that this conclusion needs to be revised.

My main concern is related to its interpretation and degree of novelty. The authors build their argumentation on the fact that the Machida-Shibata bound state has not been detected. However, giving an unusual name to a bound state does not make it new! Though it is true that Machida and Shibata were the first to perform a theoretical calculation for a resonant level in a superconductor, bound states related to (interacting) resonant levels have been studied both theoretically and experimentally in numerous mesoscopic systems. For example there have been numerous papers studying various quantum dots systems in proximity of superconducting leads. I am quoting only these two: JD Pillet et al., Nature Physics 6, 965 (2010); T. Dirks, Nature Physics 7, 386 (2011) but many have

followed. The main difference compared to the present manuscript is the presence of local Coulomb interactions (i.e. a finite U Anderson model is used to interpret the data) and the bound states studied in the aforementioned papers carry the more conventional name of Andreev bound states. In this sense, the Machida-Shibata bound states can be regarded as a particular case of Andreev bound states. This is not only a question of semantics. It is legitimate to ask how the present work compare to previous studies about Andreev bound states in various quantum dots? In particular, the authors should state what are the new physical features (and there are some) the present study bring comparatively.

We thank the Reviewer for pointing out this opportunity to optimize our manuscript and to further highlight the novelty of our findings. Beyond the following discussion about the nature of the MSSs, we would like to highlight at this point that a significant section of our work's novelty lies in the understanding of the proximity effect induced into a single eigenmode of a quantum corral which, as also explicitly acknowledged by Reviewer #2, is novel in itself.

We agree with the Reviewer that there are in fact similarities between the Andreev bound states (ABS) found in transport experiments on quantum-dot-superconductor-hybrids and the MSSs observed here. Further, we agree that the terminology of what is referred to as ABSs is somewhat broad these days. Originally, the term "ABS" describes bound states in S-N-S or S-S'-S junctions. However, various researchers have also dubbed Yu-Shiba-Rusinov states in spinful quantum dots, Majorana-zero-modes, or Caroli-de-Gennes-Matricon states as well ABS (see, e.g., Sauls, Phil. Trans. R. Soc. A 376 (2018), Shan *et al.*, Nat. Phys. 7, 325-331 (2011) or De Franceschi *et al.* Nat. Nano. 5, 703-711 (2010)) although these are all states with distinct physical properties realized in different platforms. In this sense, the term "ABS" has been used as a more general hypernym and consequently, the Reviewer is right that one can refer to the MSSs as a special type of ABS. In the following, we will summarize the special properties of the MSS found in our work, in which sense they are distinct from previous observations and how they compare to transport studies of quantum dot (QD)-superconductor (SC) hybrids.

A lot of research on ABSs in QD-SC hybrids focuses on spinful QDs in the Coulomb blockaded regime where single electrons can be added and removed from the dot by a gate voltage. Here, the Coulomb charging energy (U) competes with the superconducting pairing (Δ_s) (see, e.g., the paper by Dirks *et al.* Nat. Phys. 7, 386-390 (2011) cited by the Reviewer). In our study, the effect of U can be fully neglected because of the charge screening of the metallic bath surrounding the QD. Furthermore, our QD states have relatively narrow widths Γ indicating a coupling to the SC on the order of Δ_s , which is smaller than the energetical separation δE_r between the QD states as tuned by the QD size. Thus, we are in a regime where $U \ll \Delta_s \approx \Gamma \ll \delta E_r$. As a consequence, the QD levels are always spin-degenerate. Note that previous works on carbon nanotubes coupled to superconductors were found to have negligible Coulomb energy U ; however there, $\Gamma/\Delta_s \approx 10$ and no MSSs are expected to be observable. In experiments like the ones by Pillet and Dirks *et al.* cited by the Reviewer, it is concluded that Coulomb interactions were important for the formation of sub-gap states. Still, MSSs appear even in the complete absence of Coulomb interactions U . This regime is hard to establish in semiconducting materials where small coupling to the leads naturally leads to large Coulomb interactions on the QD. Only very recently, there have been two studies of Josephson junctions (JJs) containing a QD in a similar parameter regime as the one we realized (Bargerbos *et al.*, PRL 124, 246802 (2020) and Kringhøj *et al.*, PRL 124, 246803 (2020)). There, the device is designed to have negligible charge dispersion even in the weakly coupled regime of comparably small Γ in order to enable near-unity JJ transparency which is advantageous for transmon qubit devices. In the theoretical modelling of the JJs presented in these two references, the regime studied by us is found at zero phase difference $\phi = 0$ of the JJ.

However, these complex gated devices, which intricately rely on a two-lead setup and exceed the length scales of our setup by a factor of 1000, are based on an entirely different building mechanism, suppressing the charging energy by a conducting channel with transmissions close to unity. None of these works were able to perform tunnel spectroscopy of the ABSs and the authors did not establish the connection to the first work by Machida and Shibata (their paper is not cited). Moreover, unlike in transport setups, we spatially map the wavefunction of such ABSs in the regime of negligible electron-

electron interactions in real space for the first time. To the best of our knowledge, spin-degenerate ABSs in *s*-wave superconductors (i.e. sub-gap states distinct from YSR states) have not been imaged before using real-space techniques like STM. Thus, our work goes clearly beyond previous experiments on QDs.

Further, we would like to emphasize one of the major conclusions of our manuscript: in the limit of negligible electron-electron interactions, MSSs are not pair-breaking but they could rather be understood as the coherence peak of the induced gap in the resonance level. This is a new flavor we add to the theory going beyond the interpretation by Machida and Shibata. Finally, the atom-by-atom design of our QDs makes a scaling to QD molecules or larger artificial lattices straight-forward.

We have added the above two references and modified the abstract, introduction and discussion sections of the main manuscript text at several locations (marked in red, pages 3, 6 and 10) in order to properly highlight the advancement and importance of our study with respect to previous work on ABSs in QDs.

More specific questions and comments:

i) The authors measure a proximity –induced superconducting gap of order 1.35meV in the thin-Ag islands. How homogeneous is the gap? Along the same line, is superconductivity in AgDL in the ballistic regime or diffusive one or somewhat intermediate?

The measured gap of $\Delta = 1.35$ meV in the bulk states of Ag is laterally very homogeneous over tens of nm all the way across islands of similar height (see Supplementary Fig. S9d for an example where the coherence peak of the bulk states remains at constant energy over more than 60 nm). We have collected gap spectra on islands of various thicknesses, ranging from 10 to 120 nm. We found an overall trend that the coherence peaks are moving inwards for thick islands, in agreement with a reduced proximity effect into the bulk states. However, unlike for other heterostructures such as Au/V(100) (c.f. Vaxevani *et al.*, Nano Lett. **22**, 6075-6082 (2022)), the decay of the superconducting gap size is long-range such that the gap on 120 nm thick Ag islands is still about 75% the size of the Nb(110) gap. We point out that the number of defects on the surface of our Ag(111) islands is very low (c.f. Supplementary Fig. S2a, there are two defects within 100x100nm²) and a similarly low defect density is expected in the bulk (which cannot be probed by STM). Furthermore, the fact that the DL grows pseudomorphically on Nb(110) indicates a very good interface quality. Thus, we expect the mean free path for quasiparticles in the Ag films to be several tens of nm large. Altogether, these observations indicate that superconductivity in the DL as well as in the thick islands is in the ballistic regime.

ii) Data were taken at 4.5K with a superconducting tip to increase energy resolution. Even with such a tip, I would have expected some substantial thermal broadening of the in-gap states. However the in-gap states in Fig 2 look very sharp. Is the measured width consistent with thermal broadening? Details would be welcome.

The measured linewidth of the in-gap states is about 80 μ eV, which is not in conflict with the experimental thermal broadening expected at 4.5K because the use of superconducting tips circumvents the Fermi-Dirac broadening present in typical STS experiments to a large extent. In fact, however, we were positively surprised by the sharpness of all sub-gap features in SIS tunneling and we are not aware of published data of similar quality taken at 4.5K from other labs. We do, however, point out that the SIS tunneling quality of the data in Goedecke *et al.*, ACS Nano **16**, 14066-14074 (2022) measured using the same setup is similar to the data presented here in terms of the sharpness of sub-gap peaks. Moreover, we have compared SIS tunneling spectra on clean Nb(110) taken both at 330mK and 4.2K with the same tip previously and did not find significant changes in the sharpness of all features, but only the appearance of thermal resonances (c.f. Supplementary Note 2 of our

manuscript). One possible reason for this improvement beyond previous works could be the use of bulk Nb tips. Most other works use tips with tiny superconducting clusters attached to a metallic tip via indentation into, e.g., Pb. This may result in an imperfect gap with a magnitude much smaller than the bulk gap. Therefore, the use of mechanically sharpened Nb bulk tips may be a promising route for future high-resolution STS studies without requiring ultra-low temperatures.

iii) *The authors write “The observation of in-gap states is a surprising result, since particle-hole-symmetric states inside the gap of an s-wave superconductor are commonly believed to only appear for magnetic impurities”*

This is the author’s opinion. As I wrote already, there is whole section devoted to it in the classical review by Balatsky, Vekhter, and Zhu, Rev. Mod. Phys. 78, 373, (2006)) and Andreev bound states have been largely studied in other mesoscopic systems.

We agree that these states have been theoretically discussed before. However, by this sentence, we want to emphasize the surprising experimental observation in an STM experiment. In the original work by Machida and Shibata as well as in the mentioned Rev. Mod. Phys. review, it was always concluded that they are irrelevant for realistic scenarios, which turns out to be not generally true, as we show in our study. Therefore, it is fair to say, based on the STM literature of the past 15 years, that the appearance of these states is surprising at least for this community. It is of course true that ABSs in QDs in the regime of considerable electron-electron interactions, or Yu-Shiba-Rusinov states of magnetic atoms on superconductors, have been well studied by electrical transport methods and STS, respectively. But STM studies of the Machida-Shibata-type ABSs are completely lacking so far. Beyond this, we refer to our answer to the Reviewer’s first question. In search of a compromise, we have changed the phrase in question (page 6) to: “The observation of these in-gap states in an STM experiment is a surprising result, since impurity-induced states at particle-hole-symmetric energies deep inside the gap of an s-wave superconductor are commonly believed to only appear for magnetic impurities”.

iv) *In Fig. 4, the data deviate significantly from the theoretical curve for $|\epsilon/\epsilon_{\min}| > 1.2$. Do the authors understand why? Is the theoretical model oversimple or the assumption $|\epsilon_{\min}| \sim \Delta_{\text{ind}}$ too crude?*

The latter statement is correct and we agree with the Reviewer that it was not clear enough how this theoretical curve was calculated in the previous version of the manuscript. We have accounted for this in the revised version by adding a new Supplementary Figure S12 and an extensive description to Supplementary Note 7.

The theoretical curve follows the simple textbook expression for the particle-component of a Bogoliubov quasiparticle vs. its energy (c.f. Eqs. (S26) - (S28) in Supplementary Note 7). This expression is, however, only valid in close proximity to $\pm\Delta_{\text{ind}}$ where the MSSs behave like Bogoliubov quasiparticles. We have added the new Supplementary Figure S12 showing a comparison of the theoretically more accurate energy-dependent MSS asymmetry using the Green’s function theory of Eq. (9) to the standard Bogoliubov relation to clarify this point. As a notable difference, the asymmetry of the MSSs approaches zero (θ_B approaches $\pi/4$) again as they shift towards the coherence peaks of the bulk gap. In contrast, the Bogoliubov relation saturates and becomes fully particle-like for $E \ll 0$ and fully hole-like for $E \gg 0$. For the former theory, which is valid for a vaster range of energies, we find an agreement within the error bars of the experimentally measured values of the particle-hole mixing (see Supplementary Figure S12), providing additional evidence that the model fits the data well.

v) *It is not a priori obvious that the simple Machida-Shibata (MS) model used in the analysis would*

apply to describe the bound states associated to delocalized modes of the billard. However, the complementary study given in Sec. S6 seems to confirm that the modes of the billard can be indeed proximitized. Can the authors show the consistency of their chosen parameters with the experimental ones (by comparing dimensionless ratios)?

We in fact included the tight-binding study in Supplementary Note 6 to confirm that a mode that is delocalized over several tens of nanometers experiences qualitatively the same behavior as a localized MSS. For these calculations, we did not only try to consider the same order of dimensionless ratios as in the experimental findings, but we rather tried to match or be close to the absolute values deduced from the experiment for the following parameters: lattice structure (fcc), quadratic dispersion relation of the Ag surface state (energetic position of band bottom $E_B = 55$ meV, Fermi momentum $k_F = 0.7/\text{nm}$, consistent with the experimental data of Supplementary Note 4) and the size of the quantum dot (3 nm – 20 nm). We have converted the dimensions of the dot in Supplementary Note 6 and Supplementary Figure S11 to nanometers so that it is immediately clear to the reader that we are modelling the correct geometry. As a known drawback of tight-binding models, we need to consider an artificially large superconducting gap of $\Delta_s = 25$ meV in order to avoid finite-size effects in the 150x150-sites system. To compensate this, we increase the coupling $t_{\text{SC-SS}}$ likewise, such that the dimensionless constant $\epsilon_{\text{min}}/\Delta_s \approx 0.106$ is within the parameter regime of the MSS model shown in Fig. 3a in the main manuscript. Note that this value is still considerably far away from the experimental values (ranging from 0.5 – 0.9 according to Fig. 2d of the main manuscript). Unfortunately, we cannot increase the coupling $t_{\text{SC-SS}}$ further without decreasing the effective superconducting gap of the host layer too much. This is a drawback of the superconductor being modeled as a 2D sheet. We therefore want to emphasize that the tight-binding model is only a plausibility check.

Instead, there are much simpler arguments, why the a priori delocalized billiard state acts as a localized state. To this end, we set up the MSS model for an extended impurity (delocalization length L_{imp}) along the lines of Eq. (4) to Eq. (7) of the main manuscript and reach at integrals of the form $\int dk |V(k)|^2 \frac{E - \epsilon_k}{E^2 - \epsilon_k^2 - \Delta_s^2}$, where the k -dependent $V(k)$ only differs significantly from 0 in the interval $[k_F - 1/L_{\text{imp}}, k_F + 1/L_{\text{imp}}]$. Linearizing the dispersion around the Fermi energy as in the main manuscript, we find that the results of the manuscript remain qualitatively unchanged when $L_{\text{imp}} \ll \hbar v_F/\Delta_s = \xi$, with ξ being the coherence length of the proximity-superconducting Ag bulk. Hence, physically speaking, if the quantum dot is much smaller than the coherence length of the proximity-superconducting Ag bulk, the quantum dot mode can be regarded as localized. For bulk Ag, $253 \text{ nm} < \xi < 789 \text{ nm}$ (c.f.: *Phys. Rev. B* **32**, 4969–4976 (1985)), which means we can safely neglect the extent of the quantum dot mode of up to several tens of nanometers.

We added a corresponding note on page 7 of the main manuscript and a paragraph to the Methods section of the main manuscript to discuss why the quantum dot can be regarded as localized.

vi) Furthermore, I do not see how I can reconcile the nice study of Sec. VI with the simple MS model. Can the authors establish how the parameters of the MS model be related to the model presented in S6?

The connection between the tight-binding (TB) model of Supplementary Note 6 and the Machida-Shibata (MSS) model is as follows: In the TB model, the surface layer with the quantum dot is weakly coupled to the superconducting substrate. In the low-energy regime, the level of the quantum dot that is closest to the Fermi level is the most relevant and corresponds to the spin-degenerate level in the MSS model. This level is then coupled to the superconducting host by a weak coupling $t_{\text{SC-SS}}$ and exhibits the proximity-induced superconductivity. The free parameters of the MSS model are the superconducting gap size Δ_s , the energy of the dot E_r , and the effective coupling of the impurity to the bulk Γ , which combines the relevant data from the superconducting host and the bare coupling to the impurity. Hence, the parameters of the MSS model (see main manuscript text) corresponding to

the TB model of Supplementary Note 6 are effectively compared by two dimensionless ratios. First, we consider E_r/Δ_s , which tunes from well below -1 to well above 1 , in both models, as it should, indicating that the energy of the quantum dot level shifts through the superconducting gap when the dot changes size. Second, the ratio Γ/Δ_s is conveniently extracted by the gap size of the lines in Supplementary Fig. S11c in the limit of weak coupling between the dot and the host superconductor where $\Gamma = \varepsilon_{\min}$ (see Supplementary Note 7). Therefore, as discussed in the reply to the previous question, we arrive at $\Gamma/\Delta_s = \varepsilon_{\min}/\Delta_s \approx 0.106$ (TB model).

We would like to stress again that the supplemental TB model in Supplementary Note 6 is a qualitative demonstration that a delocalized mode can experience the superconducting proximity pairing as in the MSS model. Yet, as discussed in our reply to the previous question, since our quantum dots are much smaller than the coherence length of the proximity-induced superconducting bulk Ag, the spatially delocalized level of the TB model and the localized level of the MSS model behave quantitatively the same. In order to keep the content as fundamental as possible we, therefore, focus on the MSS model for localized impurities in the main manuscript.

Referee #2 (Remarks to the Author):

Summary: Schneider and co-authors present a comprehensive study of confined states in quantum corrals and their interaction through the proximity effect with a superconducting order parameter. The manuscript is well presented and accessible, the conclusions are supported by the experimental data and further backed by extensive additional analysis and supporting material. I recommend publication of the article as-is, with no need for further revision. My assessment is based on the following considerations:

A - Key Results

To my mind, the key results of the work are:

- 1.- The experimental demonstration of sub-gap states from a spin-degenerate impurity level coupled to a superconductor*
- 2.- Their explanation through a theoretical model*
- 3.- Establishing surface-bulk scattering as the mediator for the proximity effect in surface states.*

B - Originality and Significance

As the authors note, the proximity effect is a key ingredient for designer quantum materials. The most notable example in the recent past, at least to my mind, would be the discovery of Majorana bound states in a variety of proximitised nanostructures. The authors give several further examples in the manuscript. From this point of view, a systematic study of the proximity effect from minimal ingredients is desirable and may inspire new generations of designer quantum states. In that vein, quantum corrals are perhaps the dominant building block to study artificial lattices in the STM, but have not yet been paired with superconductivity. The authors show that the confined levels do indeed produce localised states which are affected the proximity effect and could serve as building blocks for exotic states of matter. Sub-gap states in superconductors have been extensively studied by local probes. The field focuses exclusively on the Yu-Shiba-Rusinov states to the point that sub-gap states are seen as direct evidence of spinful impurity levels. The present work challenges this view rather directly and introduces a new kind of sub-gap state with its own physics. While the results of the paper are immediately relevant in the context of local probes and artificial lattices, I believe that the fundamental physics will be interesting to a much wider audience.

We thank the Reviewer for acknowledging that our observations will be of significant interest, also outside of the STM community.

C - Data and Methodology

The experiment is well designed and executed. By adjusting the size of the corral, the authors are able to tune the energy of the resonance states in their system and collect data over a wide parameter space. Data is of excellent quality, clearly presented, thoughtfully analysed, and supports the conclusions.

D - Appropriate use of statistics and treatment of uncertainties

It is not quite clear to me whether the error bars in figures 2d and 4 conform to the editorial policy of being 'defined in the corresponding figure legends', please cross-check this.

We thank the Reviewer for making us aware of this point and **we have added the definition of the error bars to the according Figure captions.**

E – Conclusions

The conclusions follow from the analysis presented in the paper. They are supported by the data and further backed by extensive additional analysis and derivations in the supporting material.

F – Improvements

As a minor note, the figures, the axis labels in particular, appear heavily pixelated in the current format when the view is stretched to the width of the screen. I assume that this is because the figures are prepared according to the guidelines for publication in single- or double-column format, but please make sure they have sufficient resolution to be easily legible in the publication format.

We thank the Reviewer for this suggestion. Indeed, the linewidths of the axis are 0.5pt and thus the minimum width allowed in the guidelines. **They will be provided as vector graphics for the final submission such that the pixel resolution should not be a problem.**

G – References

The paper contains an extensive list of references touching on all relevant topics.

H - Clarity and context

I find the paper well-presented, clear, and accessible.

I - Inflammatory material

I find no cause for concern in this article.

We thank the Reviewer for this accurate summary, the clear praise of our work's quality and the suggestion to publish it without further revisions.

Referee #3 (Remarks to the Author):

Schneider et al investigate superconductivity in quantum dots crafted on a proximitized superconducting surface. They observe states within the superconducting energy gap that shift depending on the size of the quantum corral. They interpret these states as Machida-Shibata (MS) states, which have been predicted decades ago, but were assumed to lie extremely close to the gap edge when single-atom scatterers are considered. Hence, the observation and tunability of these states is surprising and interesting. This being said, I am not yet convinced about the correct interpretation of

the origin of these states. In my opinion there is not sufficient evidence from the data that the states are not conventional Andreev bound states which can be tuned by the coupling energy, similar to what is done in conventional transport experiments using quantum dots.

We thank the Reviewer for acknowledging that our results are surprising and interesting. In the following, we will elaborate on why we are very confident about our interpretation of the sub-gap states as Machida-Shibata states (MSSs) and why our analysis is in fact sufficiently accurate. To that end, we have added the two new Supplementary Figures S5 and S6 together with an extensive description of our analysis to Supplementary Note 4 (pages 6-10), which clarify these points. Moreover, we have added a more detailed differentiation between our work on MSSs and previous studies on ABSs in the abstract, introduction, and discussion sections (pages 3, 6 and 10) of the revised manuscript and especially refer to our reply to Reviewer #1 to substantiate the novelty of our work.

A large part of the arguments for the assignment to MS states is based on the correlation of the energy of the MS states and the coupling energy of the quantum-well states to the bulk substrate. However, the analysis of the coupling energy does not seem to be appropriate in this context. The relevant coupling energy should be the one of the state crossing the Fermi level. Contrary to this, the authors deduce the coupling strength from a bunch of measurements including those of higher-lying states in their analysis.

We thank the Reviewer for pointing out this important topic. We will address this point combined with similar comments of the same Reviewer in an extended section below.

Looking at Figure 2c, I am also not convinced by the position of the white dashed lines. The data in the background seems to be much flatter than the indicated L_x^{-2} behavior. Additionally, the lines do not seem to cross the gap at the same time as the minimum energy of the subgap state is reached. Possibly, this is just not clear from the data representation, and can be resolved by plotting the individual spectra in a stack (at least as Extended Data Figure) and showing the peak heights and shifts more explicitly.

We thank the Reviewer for pointing out this oversight in our data representation. We have noticed that there was a slight inaccuracy in our fitting of the resonance energies $E_r(L_x)$ in Supplementary Fig. S7c which we apologize for: as we are measuring with a superconducting tip, the measured energies are not expected to follow a L_x^{-2} trend as previously stated, but all energies should instead be shifted outwards by $\Delta_{\text{tip}} = 1.35$ meV. We have corrected for this error in the revised version of the manuscript, in particular Fig. 2c, and in Supplementary Figure S10. With these changes implemented, it is now obvious that the dashed lines in Fig. 2c meet at zero energy and that the MSS energy is lowest at $E_r = 0$. Note, however, that this error in the previous version did not affect the conclusions of our work but is merely related to the guide to the eye drawn in Fig. 2c and to the calculated particle-in-a-box modes in Fig. 1e and Fig. 2b. The latter have been recalculated for the new parameters, but do not exhibit clear changes compared to the previous version.

Indeed, it is very hard to clearly illustrate the trend in the features outside the gap together with the sub-gap states in one color code because of their strongly different intensities in the raw data (it is easier to see in Supplementary Fig. S10d, though). Thus, as suggested by the Reviewer, we have added the new Supplementary Figures S5 and S6 showing the full raw data measured on 34 different QDs used for creating Fig. 2c. In particular, these additional figures make it much more clearly visible how the eigenmodes outside the gap shift through E_F . We believe that the trends are now very clear from these new Supplementary Figures and that the white dashed lines in Fig. 2c are thus trustworthy as a guide to the eye.

Another problem is that the width of the quantum well states seems to be much broader than shown in Figure 2d. It occurs to me as if the peaks are at least 5mV in width independent of the quantum number n_x . Again, a more detailed presentation of the data may help. I am aware that Supplementary Note 4 addresses the determination of the width, but it also only shows the data of one corral in Figure S5a-c, which is even an example, where there is no resonance crossing the Fermi level and the width of $n=3$ is broader than $n=2$, contrary to the extracted value of the dashed line in Fig. S5d. The authors argue that this is due to the finite lifetime, which is correct but this problem could be avoided by only considering the quantum well states at the Fermi level.

Overall, it is not clearly specified from which corrals the data in Figure S5 d is derived and which states are taken into account (see comment above). The data scatters quite significantly and it is not clear why the authors draw the constant dashed lines. There could possibly be a dependence of Γ on E instead of being constant. As Γ is one of the most important parameters, this analysis should be substantiated.

We appreciate the detailed analysis of the Reviewer. The Reviewer is bringing up a very relevant point, and we agree that the determination of Γ was not laid out convincingly enough in our previous version of the manuscript. We have thus extended this discussion in the revised version of Supplementary Note 4 to clarify our analysis.

First and foremost: the data in Supplementary Figure S7d (formerly Supplementary Fig. S5d) is derived from all corrals we have built during this series of measurements. The same data set is used to create Fig. 2c, as mentioned in the beginning of Supplementary Note 4. For clarity and transparency, we now show the raw, unedited data measured on all of these corrals in the new Supplementary Figure S5. Moreover, we have added a new Supplementary Figure S6 in order to substantiate the analysis of the eigenmodes outside the superconducting gap and to make it more comprehensible. In particular, we believe that it is now clearly evident from these two new figures that our eigenmode line widths are in fact on the order of 2 - 5 meV (FWHM), in good agreement with the analysis in Fig. 2d. In the following, we want to elaborate on why we chose to perform the linewidth analysis the way we did.

We agree that, in principle, and in sight of the theoretical description, the width of the QD level should be analyzed at the exact point where it is crossing the Fermi level. However, the Fermi level becomes gapped in the superconducting state of the sample. Thus, the line shape of the level is not only a simple Lorentzian with a well-defined width but it is complicated by the presence of the tip's and sample's superconducting gaps forming around E_F (c.f. for example Supplementary Fig. S8a or Supplementary Fig. S5 where a resonance E_r is close to E_F). Therefore, Γ cannot be determined directly from our experiments whenever $E_r \approx E_F$.

For this reason, we have to analyze the QD state's widths a few meV away from E_F and extrapolate this width towards E_F (c.f. Supplementary Fig. S7d). Here, we neglect the line width fits for $|E_r| < 5$ meV since the line shapes for these eigenmodes are clearly deformed by the superconducting gaps and are not of Lorentzian shape any more. We agree that this whole procedure adds some error to the extracted linewidths of features exactly at E_F , which we openly discuss in Supplementary Note 4, and we have put some additional focus on this issue in the revised version (page 10 of the Supplementary Information). What can be seen in Supplementary Fig. S7d is that taking the weighted average of all data points gives a result dominated by the data points with the energy closest to E_F . We therefore consider this the best *a priori* estimate of the states' broadening at the Fermi level that we can extract from the data.

Nevertheless, we do admit that there is a large variance in the linewidths as shown in Supplementary Figs. S7d and S10e and, that taking the weighted average is only a first approximation. In fact, the Reviewer is right in stating that we *do* expect an energy dependence $\Gamma(E)$. The linewidths for levels above E_F are larger in general, as can be seen in Supplementary Fig. S7d. Thus, the n_x -dependence of

Γ is obscured by its additional energy-dependence and it may occur that the $n_x = 3$ mode is broader than the $n_x = 2$ mode for a single QD length as in the example of Supplementary Fig. S7b. Nevertheless, for *constant* energies in Supplementary Fig. S7d, a clear trend towards narrower linewidths can be seen for larger n_x (the same holds for Supplementary Fig. S10e). This is in agreement with multiple previous studies on confined surface states and thus a well-known trend. For instance, as we discuss in Supplementary Note 4, it has been characterized in the past that the eigenmodes' widths increase with energies away from E_F because of enhanced coupling to phonon modes setting in at around ± 14 meV. Therefore, one does not expect a constant width of the features versus energy but an energy-dependence of Γ . However, accurately extrapolating the measured widths to the widths of the resonances at E_F requires a more elaborate model of all physical broadening processes. Choosing phenomenological fit functions would not be well-motivated; instead an analytical model for the energy dependence $\Gamma(E)$ would need to be developed, which goes beyond the scope of the present manuscript.

We want to emphasize that, even without our detailed analysis shown in Supplementary Figs. S7d and S10e, the decrease in Γ with corral size can be already seen in **the raw data now shown in the new Supplementary Fig. S5** (compare for example the apparent widths of the $n_x = 1$, $n_x = 2$ and the $n_x = 3$ state when they are at similar energies E_r). Further, the trend in ε_{\min} is also visible in the (raw!) data of Fig. 2c. Thus, the correlation between ε_{\min} and Γ shown in Fig. 2d is present even if there was a larger error in determining Γ .

The comparison of experiment and theory is not laid out clearly. Instead of plotting three graphs of different coupling strengths a direct comparison to experiment would better be compiled in one graph showing the MS state as a function of coupling strength.

We agree that a direct comparison between theory and experiment would be favorable. Let us explain the accompanying problems in our manuscript and why we chose the current presentation.

First, we believe that the point raised by the Reviewer partially stems from a misunderstanding. The suggested panel showing the energy of the MSS as a function of coupling strength is in fact already plotted in Figure 2d as the gray dashed line. **We now clearly stated in the figure caption that this curve belongs to the Green's function theory of Eq. (9).** Concerning the separation of the experimental results in Fig. 2c and the theoretical results in Fig. 3, we chose this representation in order to stick to the raw data as closely as possible. Note that the experimental data is dI/dV data obtained with a superconducting tip and in dependence on the length of the quantum dot, while the theoretical data is the LDOS in dependence on the energy of the quantum dot. For this reason, our experimental Fig. 2c basically shows the effect of tuning the resonance level's energy E_r across E_F . However, the shift in E_r is reversed in comparison to the theory Figure 3 since the eigenmodes move *down* in energy as the corral size *increases*. To merge the experimental and the theory figure, a deconvolution of the data to account for the tip's gap and a conversion of the length of the quantum dots into quantum dot energies would be necessary. As we believe that this procedure would obstruct the reader's view and that reversing the E_r axis just for the sake of a better 1:1 correspondence with the experiment would be artificial, we would prefer to keep the comparison between experiment and theory separate in Fig.2c and Fig.3.

Instead, in order to further clarify the behavior of the MSSs as parameters of the model are varied, and to make the experimental and theoretical results more easily comparable, **we have added the new Extended Data Figures 1 and 2.** Extended Data Figure 1 shows the energetic position of the in-gap states with the coupling strength Γ and E_r as requested by the Reviewer. Extended Data Figure 2 enables a direct comparison of the experimental and theoretical data for quantum dots of three different lengths, which we further describe in our answer to the next comment of the Reviewer.

Additionally, in experiment the minimal energies are plotted as a function of coupling strength (Fig.2d). An additional plot of the MS states as a function of energy of the QWS would be helpful for direct evidence of the avoided crossing and direct comparison to Fig.3. Such representation may be better suited to show the avoided crossing.

We thank the Reviewer for this suggestion. A major problem with such a representation is the fact described above that we cannot determine the quantum dot level energy E_r from a fit when the level is at energies below 3-4 meV. Note, however, that the change of E_r with L_x is almost linear in the narrow energy range of ± 4 meV around the gap. Therefore, Fig. 2c shows approximately the measurement proposed by the Reviewer: the MSS energy as a function of the quantum dot level energy E_r illustrated by the white dashed lines. Thus, a qualitative comparison between Fig. 2c and Fig. 3 is already straightforward, while a conversion of the vertical axis in Fig. 2c from L_x to E_r is expected to mainly introduce additional systematic errors. Nevertheless, **we have added this analysis and a direct comparison to the theory to the new Extended Data Figure 2.** We plot the same spectra from Fig. 2c versus the anticipated resonance energy E_r extracted from the fitted white dashed lines describing the evolution of $E_r(L_x)$. The overall agreement of the experimental data and the theory panels is decent. Note, that there are subtle differences between the experimental and theory panels i) because the dI/dV experimental data obtained with a superconducting tip is not deconvoluted and ii) because only the LDOS of the quantum dot level is considered in the model whereas the experimental tunneling occurs both into the quantum dot level and into the substrate host superconductor. For this reason, the substrate coherence peak at Δ_s is absent in the simulation unlike in the experimental data. While one could in principle account for this by calculating the total LDOS, the ratio of tunneling into the quantum dot level vs. tunneling into the substrate states is not known a priori. Therefore, calculating the total LDOS would also add another free parameter and we decided against this.

We hope that we were able to convince the Reviewer that our analysis is well-conceived and that our interpretation of the experimentally observed in-gap states as MSSs is thus solid. We are confident that the revised versions of our manuscript and Supplementary Information are now suitable for publication.

Reviewer Reports on the First Revision:

Referees' comments:

Referee #1 (Remarks to the Author):

The authors have satisfactorily answered the questions and comments I and referee 3 raised. The authors have therefore modified the manuscript accordingly and added some relevant explanations and materials in the Supplementary Material. I do not have any further comments and recommend the manuscript for publication.

Referee #3 (Remarks to the Author):

The authors made strong effort in their response to the reports and added data to the Supplementary Information. My previous concerns have been solved by the additional data. However, Figure S7 is still confusing. The state with $n=2$ appears narrower than the state with $n=3$. Despite of this graph, their statistical analysis seems to support the opposite trend with increasing n . They argue why this discrepancy may be there. Yet, I would strongly suggest to exchange this panel against a consistent example.

Author Rebuttals to First Revision:

Yellow marked text = here we added or changed something in the manuscript/supplement
The corresponding changes appear as **red text** in the manuscript/supplement

Referee #1 (Remarks to the Author):

The authors have satisfactorily answered the questions and comments I and referee 3 raised. The authors have therefore modified the manuscript accordingly and added some relevant explanations and materials in the Supplementary Material. I do not have any further comments and recommend the manuscript for publication.

We are glad that all the Reviewer's questions were answered and sincerely thank for the recommendation to publish our work as is.

Referee #3 (Remarks to the Author):

The authors made strong effort in their response to the reports and added data to the Supplementary Information. My previous concerns have been solved by the additional data. However, Figure S7 is still confusing. The state with $n=2$ appears narrower than the state with $n=3$. Despite of this graph, their statistical analysis seems to support the opposite trend with increasing n . They argue why this discrepancy may be there. Yet, I would strongly suggest to exchange this panel against a consistent example.

We thank the Reviewer for this recommendation. Indeed, the energy dependence of the linewidth dominates in this example, which might be confusing to the reader. **We have accordingly exchanged the panels in Supplementary Figure 3 (formerly Fig. S7) and now show three examples from Extended Data Fig. 1 where the eigenmode energy is similar.** As a result, the $n_x = 1$ mode is indeed the broadest, followed by the $n_x = 2$ and the $n_x = 3$ mode. We hope that this satisfactorily solves the problem.